# Isolated catatonia-like executive dysfunction in mice with forebrain-specific loss of myelin integrity

Sahab Arinrad[1†], Constanze Depp[2†], Sophie B Siems[2], Andrew Octavian Sasmita[2], Maria A Eichel[2], Anja Ronnenberg[1], Kurt Hammerschmidt[3], Katja A Lüders[2], Hauke B Werner[2], Hannelore Ehrenreich[1*], Klaus-Armin Nave[2*]

[1]Clinical Neuroscience, Max Planck Institute for Multidisciplinary Sciences, Göttingen, Germany; [2]Neurogenetics, Max Planck Institute for Multidisciplinary Sciences, Göttingen, Germany; [3]Cognitive Ethology, German Primate Center, Göttingen, Germany

**Abstract** A key feature of advanced brain aging includes structural defects of intracortical myelin that are associated with secondary neuroinflammation. A similar pathology is seen in specific myelin mutant mice that model 'advanced brain aging' and exhibit a range of behavioral abnormalities. However, the cognitive assessment of these mutants is problematic because myelin-dependent motor-sensory functions are required for quantitative behavioral readouts. To better understand the role of cortical myelin integrity for higher brain functions, we generated mice lacking *Plp1*, encoding the major integral myelin membrane protein, selectively in ventricular zone stem cells of the mouse forebrain. In contrast to conventional *Plp1* null mutants, subtle myelin defects were restricted to the cortex, hippocampus, and underlying callosal tracts. Moreover, forebrain-specific *Plp1* mutants exhibited no defects of basic motor-sensory performance at any age tested. Surprisingly, several behavioral alterations reported for conventional *Plp1* null mice (Gould et al., 2018) were absent and even social interactions appeared normal. However, with novel behavioral paradigms, we determined catatonia-like symptoms and isolated executive dysfunction in both genders. This suggests that loss of myelin integrity has an impact on cortical connectivity and underlies specific defects of executive function. These observations are likewise relevant for human neuropsychiatric conditions and other myelin-related diseases.

**\*For correspondence:**
ehrenreich@mpinat.mpg.de (HE);
nave@mpinat.mpg.de (K-AN)

[†]These authors contributed equally to this work

## Editor's evaluation

This paper shows that conditional knockout mice develop wide areas of loss of myelin integrity, as well as microglial inflammation within the affected areas. In association, a range of behavioral tests reveal abnormalities as the mice age. This work presents an important link between myelin loss and aging that will be of interest to a broad swathe of the scientific community.

## Introduction

In the central nervous system (CNS), oligodendrocytes synthesize myelin to facilitate rapid impulse conduction of axons and to support their functional integrity (*Nave and Werner, 2014*; *Stassart et al., 2018*). The formation and preservation of healthy myelin is a prerequisite for normal motor and sensory functions, as indicated by human myelin diseases and their corresponding mouse models, which have been studied in more detail at the molecular and cellular level. The severe developmental defects of myelination that affect children at a young age are dominated by the lack of motor

development and severe neurological defects, often leading to premature death. Only milder perturbations of myelination make the associated delays of cognitive development obvious, but they are still masked by severe neurological impairments. Some demyelinating diseases of adult onset are degenerative in nature and begin with psychiatric symptoms, including psychosis, such as in metachromatic leukodystrophy (*Baumann et al., 2002*). Indeed, there is increasing evidence of oligodendrocyte and white matter defects in a range of neuropsychiatric diseases (*Nave and Ehrenreich, 2014*; *Zhou et al., 2021*). Moreover, advanced brain aging in healthy individuals is associated with subtle myelin defects, as first demonstrated by electron microscopy in aged non-human primates (*Peters and Sethares, 2002*). This raises the question whether structural myelin abnormalities play a role in age-dependent cognitive decline.

In psychiatric patients, it is difficult to decide whether the correlation of MRI detectable white matter abnormalities or the ultrastructural loss of myelin integrity are the cause or effect of (age-dependent) neuronal dysfunctions, or even caused by long-term pharmacological interventions (*Nave and Ehrenreich, 2014*). Thus, novel genetic animal models are required to define the role of oligodendrocytes and subtle myelin defects as an underlying cause of a psychiatric phenotype (*Hagemeyer et al., 2012*). Unfortunately, the cognitive testing of mice depends on quantitative behavioral readouts which are affected by subcortical and myelin-dependent motor-sensory functions.

An entire range of myelin abnormalities with a corresponding wide spectrum of clinical defects characterizes mutations of the proteolipid protein gene (*PLP1/Plp1*). The X-linked gene is highly expressed in oligodendrocytes and encodes a tetraspan membrane protein of CNS myelin (PLP) along with a minor splice isoform, termed DM20 (*Jahn et al., 2009*; *Milner et al., 1985*; *Nave et al., 1987*). At one end of the disease spectrum, *PLP1* mutations cause the expression of misfolded proteins that trigger oligodendroglial ER stress, apoptosis, hypomyelination, and connatal leukodystrophy (Pelizaeus-Merzbacher disease) with premature death in children and corresponding mouse mutants (*Nave and Boespflug-Tanguy, 1996*). At the other end, the mere loss-of-function (null) mutations of *PLP1*, such as by a genomic deletion or in *Plp1* null mice (*Klugmann et al., 1997*), cause late onset neurodegeneration (*Griffiths et al., 1998*), modeling spastic paraplegia type 2 (SPG2) in humans (*Inoue, 2019*; *Saugier-Veber et al., 1994*). The isolated loss of axonal integrity in the presence of myelin led to the concept that oligodendrocytes not only facilitate impulse propagation but also maintain the axonal integrity (*Griffiths et al., 1998*; *Nave, 2010*). The latter includes glycolytic support of axonal energy metabolism (*Fünfschilling et al., 2012*; *Lee et al., 2012*) and most likely other cellular interactions, such as anti-oxidant defense (*Mukherjee et al., 2020*) via secretion of extracellular vesicles (*Frühbeis et al., 2020*).

Loss of myelin integrity and axonal degeneration are invariably associated with secondary neuroinflammation (*Kassmann et al., 2007*; *Lappe-Siefke et al., 2003*). Recently, we discovered in different myelin mutant mice that microgliosis causes a unique behavioral phenotype: when mice were placed onto a horizontal bar with their forepaws, wildtype (WT) animals turn away within 0.2 s, whereas these mutants exhibit longer response times. We hypothesized that this delay reflects the loss of higher brain (executive) functions rather than a motor impairment. Strikingly, these 'catatonic signs' were completely lost following the targeted pharmacological depletion of microglia (*Janova et al., 2018*). In an independent study (*Gould et al., 2018*), *Plp1* null mice showed defects in the 'puzzle box' paradigm, which measures the escape latency from a brightly lit open space into a shelter via a mechanically blocked entry. Successfully overcoming these obstacles is also considered an 'executive function' (*Ben Abdallah et al., 2011*; *Gould et al., 2018*; *O'Connor et al., 2014*; *Pease-Raissi and Chan, 2018*).

Human catatonia and the loss of executive functions may result from defects in the frontal lobes and its underlying white matter (*Arora and Praharaj, 2007*; *Northoff et al., 2004*), but experimental evidence is lacking. To test whether executive functions are specifically dependent on forebrain integrity, we targeted the floxed *Plp1* allele (*Lüders et al., 2017*) in mice expressing Cre under control of the *Emx1* promoter in ventricular zone stem cells giving rise to neurons and glial cells in the forebrain (*Gorski et al., 2002*). When applying a large battery of behavioral tests on adult mutants of both sexes, we found impaired executive functions at adult ages, coinciding with well-known histopathological signs of advanced white matter aging. Surprisingly, this was in the presence of unchanged motor-sensory performance, memory functions, and social behavior. Our genetic data are thus in

strong support of the hypothesis that the integrity of myelinated fibers in the forebrain is critical for isolated aspects of executive functions.

## Results

### Targeting PLP expression in the mouse forebrain

We crossbred mice harboring a floxed *Plp1* allele (*Lüders et al., 2017*) with mice expressing Cre recombinase under control of the *Emx1* promoter (*Emx^IRESCre* mice; *Gorski et al., 2002*), yielding experimental male (*Plp^flox/Y^*Emx^IRESCre*) and female (*Plp^flox/flox^*Emx^IRESCre*) mice, termed forebrain PLP cKO in the following, as well as male *Plp^flox/Y* and female *Plp^flox/flox* mice as controls (controls) (*Figure 1A*). EMX1 is a homeobox transcription factor in ventricular zone stem cells of the forebrain, including the neocortex, corpus callosum, and hippocampal formation (*Gorski et al., 2002*).

We first validated the successful genetic targeting of oligodendrocytes in the forebrain by probing coronal brain slices with an anti-PLP antibody (*Figure 1B*). We detected substantial loss of PLP-immunopositive cells in the corpus callosum and in cortical areas (*Figure 1B'*). We validated these findings by performing light-sheet microscopy (LSM) in thick (1 mm) sagittal brain slices. Also here, only sparse PLP-immunolabeling was detected in cortical areas, the corpus callosum, and hippocampus. Expectedly, in the striatum which does not belong to the *Emx^IRESCre* recombination territory (*Gorski et al., 2002*), we did not find a considerable reduction of PLP staining. Also, the mid- and hindbrain as well as the cerebellum showed normal PLP labeling, comparable to control brains (*Figure 1C*).

The detection of a small number of PLP-positive cells in the forebrain could indicate that recombination efficacy did not reach 100%. Alternatively, WT OPCs/oligodendrocytes from adjacent non-recombined regions could have migrated to these areas. Indeed, we observed a ventral-dorsal gradient of PLP-immunopositivity in the corpus callosum with ventral axons that directly overlie the non-recombined striatum having considerable amounts of PLP+myelin (*Figure 1C* closeup).

Confirming our image-based analysis, PLP/DM20 was hardly detectable by immunoblots of prefrontal cortical lysates (*Figure 1D*). The abundance of myelin basic protein (MBP), another myelin-specific structural protein (*Boggs, 2006*; *Nawaz et al., 2013*), was unchanged (*Figure 1D*). Using the cerebellum as a control region that lacks Emx-Cre expression, the abundance of PLP/DM20 did, again, not differ between genotypes (*Figure 1D*).

We next investigated myelin integrity in the forebrain in PLP conditional knockout (cKO) mice by performing MBP and PLP colabeling (*Figure 1—figure supplement 1*). While on first sight overall MBP labeling appeared similar between PLP cKO and control animals in both recombined and non-recombined areas (*Figure 1—figure supplement 1A*), close inspection of the upper cortical layers in the medial forebrain of PLP cKO mice revealed a loss of MBP+fibers (*Figure 1—figure supplement 1B*). Quantifications confirmed this local loss of myelin profiles (*Figure 1—figure supplement 1C*).

### Forebrain PLP cKO mice show restricted neuropathology and inflammation in forebrain white matter tracts

Conventional *Plp1* null mutant mice that lack PLP expression in all cells show widespread micro- and astrogliosis (*de Monasterio-Schrader et al., 2013*; *Griffiths et al., 1998*), presumably triggered by progressive axonopathy and/or oligodendroglial stress signals. We investigated PLP cKO and control brains for signs of microgliosis and astrogliosis by IBA1 and GFAP fluorescent immunolabeling, respectively (*Figure 2A and B*). We observed significant increases in the area immunopositive for IBA1 and GFAP in the corpus callosum as the primary white matter tract targeted by *Emx^IRESCre* but not in the striatum as a non-recombined white matter region. Surprisingly, cortical areas that were efficiently recombined in *Emx^IRESCre* animals did not show robust gliosis. We validated these findings by performing classical 3,3'-diaminobenzidine-immunohistochemistry for IBA1, MAC3, and GFAP in the fimbria, corpus callosum, and prefrontal cortex (*Figure 2—figure supplement 1*). Again, we observed significant gliosis in forebrain white matter tracts (hippocampal fimbria, corpus callosum) but not the cortical gray matter. In this sex-mixed cohort, we did not find obvious differences in the extent of gliosis between male and female forebrain PLP cKO mice (sex indicated by different symbols in bar graphs in *Figure 2—figure supplement 1*). We conclude that in the forebrain of PLP cKO mice gliosis is largely restricted to the targeted white matter regions.

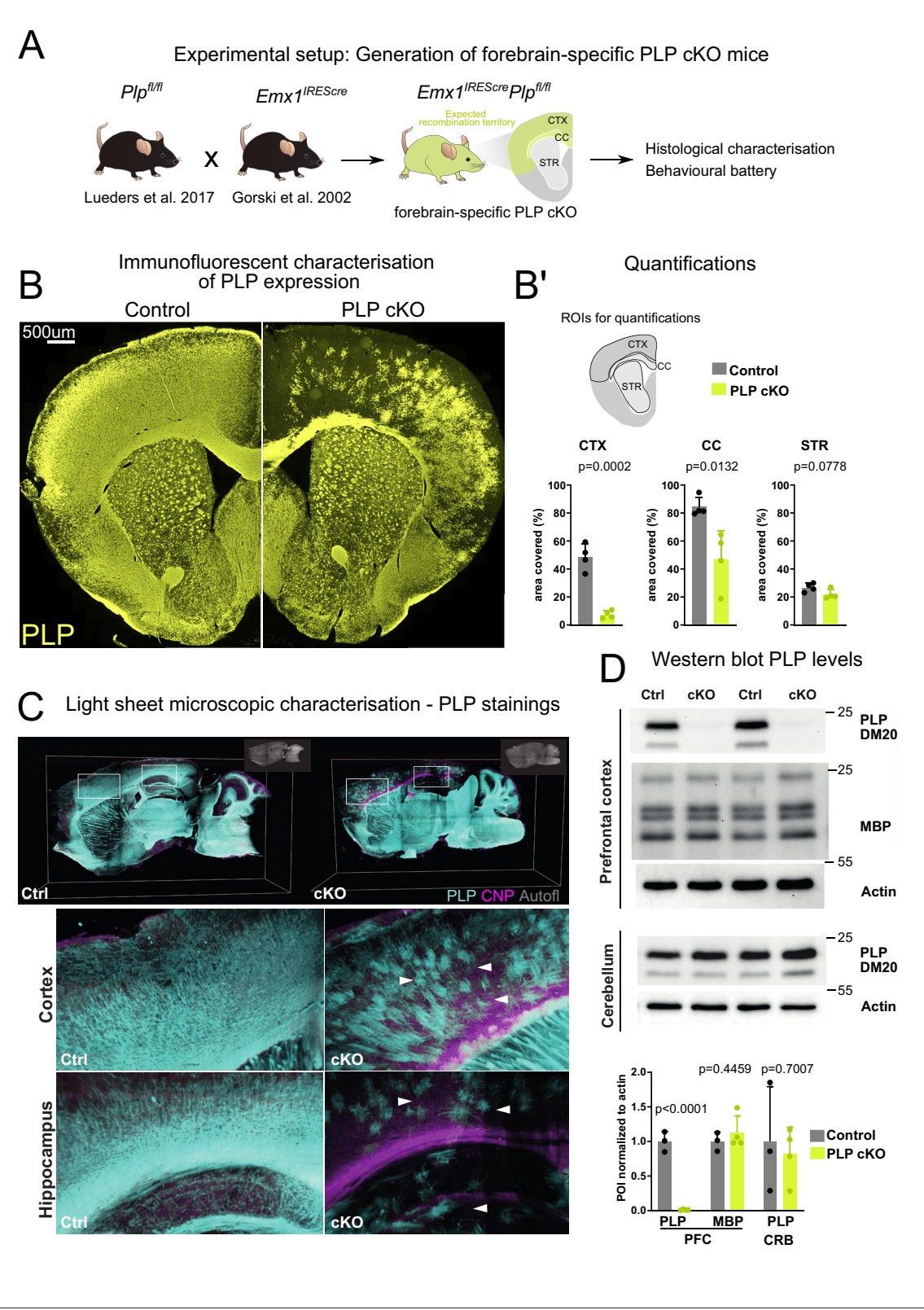

**Figure 1.** Targeting proteolipid protein (PLP) expression in the forebrain of mice. (**A**) Scheme illustrating the generation of mice with forebrain-specific loss of PLP (PLP conditional knockout [cKO]). Forebrain specificity is mediated by the *Emx1*<sup>IREScre</sup> driver line. Note that these mice are myelinated despite the absence of PLP from most cortical oligodendrocytes. (**B**) Immunofluorescence of PLP stained in the forebrain of PLP cKO mice and controls. (**B′**) Quantification of the PLP-immunopositive areas in the cortex (CTX), corpus callosum (CC), and striatum (STR). For statistical analysis, a two-sided

*Figure 1 continued*

Student's t-test was performed. Datapoints represent individual mice (**n**). n=4 for controls and n=4 for PLP cKO. Age of animals 11 months. (**C**) Light-sheet microscopy to detect PLP in forebrain PLP cKO mice. Whole-mount brain sections (1 mm) were subjected to immunostaining and clearing. Boxed areas are enlarged in the lower panels. White arrows indicate sparse PLP-immunolabeling in forebrain cortical structures and the hippocampus of PLP cKO in comparison to control mice. Autofl: autofluorescence. Age of the animals is 3 months. (**D**) Immunoblot detection of PLP and its smaller isoform DM20, myelin basic protein (MBP), and actin in lysates of the prefrontal cortex and cerebellum of forebrain PLP cKO mice. Abundance of the protein of interest (POI) was normalized to actin. Respective quantifications are shown in the lower panel. For statistical analysis, a two-sided Student's t-test was performed. Datapoints represent individual mice (**n**). n=3 for Ctrl and n=4 for PLP cKO. Age of the animals was 4 months. All data are shown as means ± SEM. Levels of significance were defined as p>0.05 (*), p>0.01 (**) and p>0.001 (***).

The online version of this article includes the following source data and figure supplement(s) for figure 1:

**Source data 1.** Uncropped/full western blots for western blot bands shown in *Figure 1D*.

**Figure supplement 1.** Loss of myelin integrity in recombined areas in forebrain proteolipid protein (PLP) conditional knockout (cKO) mice.

Conventional *Plp1* null mutant mice also develop a secondary axonopathy throughout the CNS (*Griffiths et al., 1998*). In forebrain PLP cKO mice this neuropathology was also largely restricted to the targeted white matter areas: We assessed 22-month-old mice by immunohistochemistry and found amyloid precursor protein (APP) immunopositive axonal spheroids in the hippocampal fimbria and corpus callosum (*Figure 2C*, *Figure 2—figure supplement 1*). At this age, we also noticed a pronounced atrophy of the fimbria and ventricular enlargement in PLP cKO mice – a common concomitant of neuroinflammation (*Lepore et al., 2013*; *Figure 2C*). In contrast, the prefrontal cortex itself did not exhibit an increase in APP[+] axonal spheroids in 22 months' PLP cKO mice, the oldest age tested (*Figure 2—figure supplement 1*). However, axonal pathology included some white matter structures outside the *Emx1* expression domain. We presume these reflect distal axonal projections from neurons in the target region, such as fibers in the internal capsule (*Figure 2C'*).

## Behavioral defects of forebrain PLP cKO mice

To study the behavioral consequences of forebrain specific loss of myelin integrity, we applied a large battery of behavioral tests on PLP cKO mice of both sexes at several timepoints (*Figure 3*, *Table 1*). This battery comprised tests for monitoring the overall health status, motor function and coordination, hearing, vision, heat/pain perception, sensorimotor gating, social behavior, depression-like behavior, cognitive performance in the IntelliCage, spatial memory and reversal in the Morris water maze (MWM), exploration and working memory, catatonic signs and executive functions.

In contrast to conventional *Plp1* null mice (*Griffiths et al., 1998*), PLP forebrain-specific mutants remained free of motor impairments at any age. On the rotarod, the latency to fall did not differ between cKO and control mice at 6 and 17 months of age, also when quantified separately for males and females (*Figure 3—figure supplement 1*). Interestingly, in several other behavioral paradigms, in which conventional *Plp1* null mice were affected (*Gould et al., 2018*; *Petit et al., 2014*), the PLP cKO mice were normal; for example, hot plate (pain), beam balance (coordination), motor performance, and olfaction (*Figure 3*).

On the other hand, forebrain PLP cKO mice showed and shared defects in paradigms that measured executive functions. Exclusively in our test setup, the 'bar test' was included and repeatedly performed in which mice were placed with their forelimbs against a horizontally held rod (*Figure 4A and B*; *Hagemeyer et al., 2012*; *Janova et al., 2018*; *Poggi et al., 2016*). Here, adult forebrain PLP cKO mice showed an extended response time, in which the apparent immobility constitutes a 'catatonia-like' feature (*Figure 4B*). Both female and male groups showed delayed response times on the bar as early as 3 months for females and 5 months for males (*Figure 4B*).

We also employed the 'puzzle box' paradigm, a problem-solving task, using skills obtained in previous stages of the test (*Ben Abdallah et al., 2011*; *Gould et al., 2018*; *O'Connor et al., 2014*). Placed into the brightly lit area of an open arena, the latency to reach entry into a 'safe box' was monitored, but made increasingly difficult by a series of camouflaging tools (*Figure 4C*). When assessed at the age of 20 months, both male and female forebrain PLP cKO mice exhibited significant delays compared to controls (*Figure 4D*).

We additionally found performance differences of forebrain PLP cKO mice in an adapted 'hurdle test' that measures 'executive function'. Mice were placed into the middle of a 120 cm round open field that was divided into squares (5 cm × 5 cm), separated from each other by 5 cm 'hurdles', but

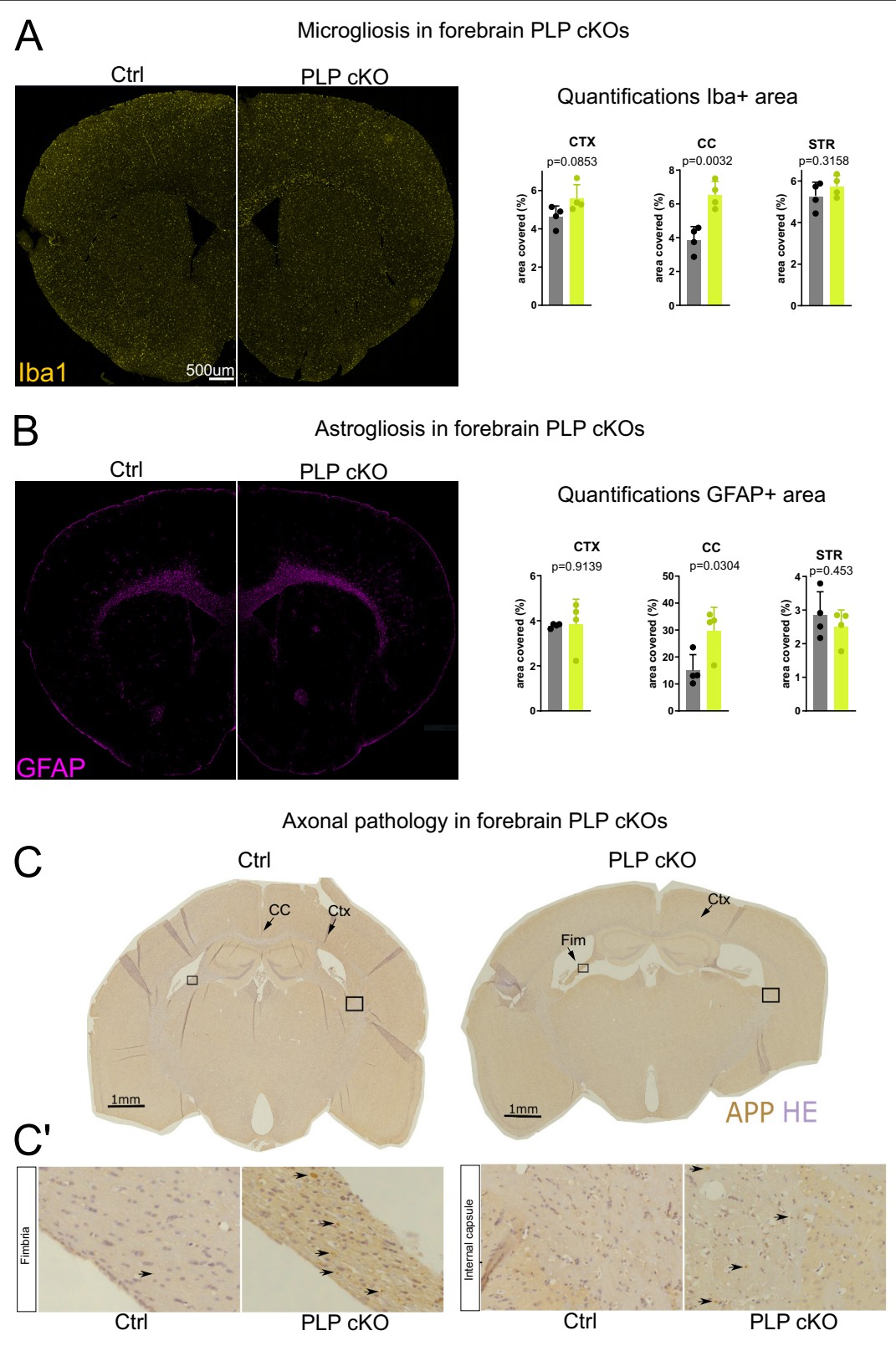

**Figure 2.** Neuropathological changes in the subcortical white matter in forebrain proteolipid protein (PLP) conditional knockout (cKO) mice. (**A**) Immunofluorescent characterization of microgliosis (IBA1 immunolabeling) in forebrain PLP cKO mice (overview coronal brain section). Quantifications for IBA1+area for region of interests (ROIs) (CTX: cortex, CC: corpus callosum, STR: striatum) are shown on the right. Bars represent means ± SEM

*Figure 2 continued on next page*

*Figure 2 continued*

and datapoints represent individual mice (n=4 per group). For statistical analyses two-sided Student's t-tests were performed. Age of the animals was 22 months. (**B**) Immunofluorescent characterization of astrogliosis (GFAP immunolabeling) in forebrain PLP cKO mice (overview coronal brain section). Quantifications for GFAP+area for ROIs (CTX: cortex, CC: corpus callosum, STR: striatum) are shown on the right. Bars represent means ± SEM and datapoints represent individual mice (n=4 per group). For statistical analyses two-sided Student's t-tests were performed. Age of the animals was 22 months. (**C**) Immunohistochemical characterization of axonal swellings (amyloid precursor protein [APP] immunolabeling) in forebrain PLP cKO mice (overview coronal brain section). Note the apparent fimbria atrophy and ventricular enlargement in forebrain PLP cKO mice. Age of analysis 22 months. (**C′**) Closeup images of boxed areas in C. Black arrows indicate axonal spheroids in the fimbria and internal capsule of cKO mice, which are essentially absent in Ctrl mice. Levels of significance were defined as p>0.05 (*), p>0.01 (**) and p>0.001 (***).

The online version of this article includes the following figure supplement(s) for figure 2:

**Figure supplement 1.** Neuropathological changes in the subcortical white matter in forebrain proteolipid protein (PLP) conditional knockout (cKO) mice.

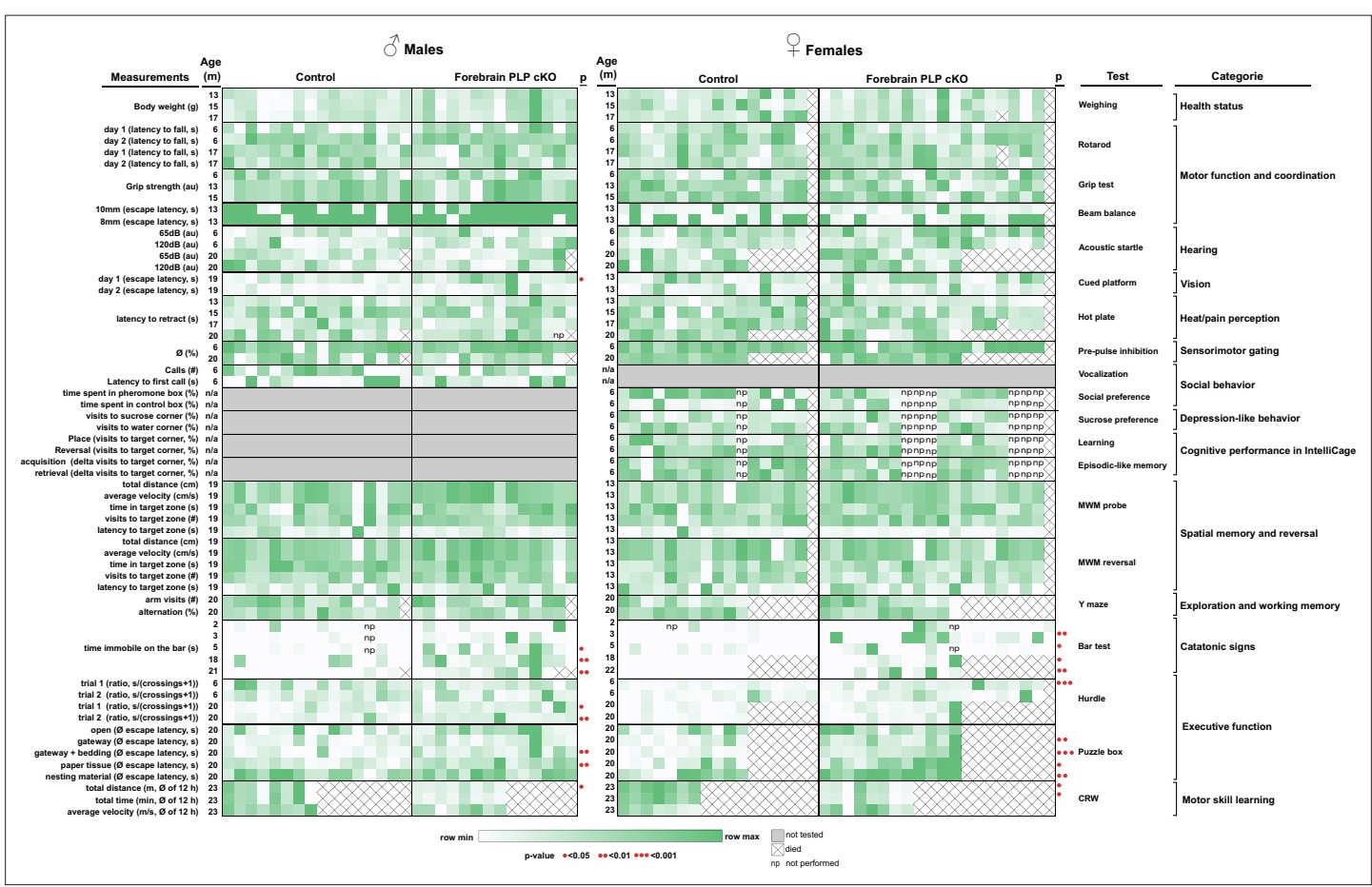

**Figure 3.** Behavioral deep-phenotyping of male and female forebrain proteolipid protein (PLP) conditional knockout (cKO) mice. Heatmap representation of behavioral data in the categories of overall health status, motor function and coordination, hearing, vision, heat/pain perception, sensorimotor gating, social behavior, depression-like behavior, cognitive performance in the IntelliCage, spatial memory and reversal in the Morris water maze (MWM), exploration and working memory, catatonic signs and executive function. Each heatmap column represents individual animals (n-numbers). Statistical significance is indicated by red dots. Raw values, n-numbers of mice tested per experiment, mean, SEM, and p-values are specified in *Table 1*.

The online version of this article includes the following figure supplement(s) for figure 3:

**Figure supplement 1.** Forebrain proteolipid protein (PLP) conditional knockout (cKO) mice display normal latency to fall from the rotarod.

**Table 1.** Behavioral deep-phenotyping of male and female forebrain proteolipid protein (PLP) conditional knockout (cKO) mice. Precise n-numbers, mean values, SEM, and p-values for all behavioral experiments are given.

| Behavioral paradigms | Males | | | | | | Females | | | | | |
| --- | --- | --- | --- | --- | --- | --- | --- | --- | --- | --- | --- | --- |
| | $Plp^{flox/Y}$ | | | $Plp^{flox/Y}$ $*Emx^{Cre}$ | | | $Plp^{flox/flox}$ | | | $Plp^{flox/flox}$ $*Emx^{Cre}$ | | |
| | Age (months) | (N) | Mean ± SEM | (N) | Mean ± SEM | p-Value | Age (months) | (N) | Mean ± SEM | (N) | Mean ± SEM | p-Value |
| **Health status** | | | | | | | | | | | | |
| Body weight (g) | 13 | 16 | 36.39±0.77 | 14 | 39.23±1.29 | 0.06 | 13 | 16 | 31.16±1.17 | 19 | 29.64±0.94 | 0.25 |
| | 15 | 16 | 37.69±0.68 | 14 | 40.51±1.31 | 0.06 | 15 | 16 | 31.52±1.24 | 19 | 30.50±1.01 | 0.52 |
| | 17 | 16 | 36.86±0.85 | 14 | 39.13±1.56 | 0.2 | 17 | 16 | 32.11±1.38 | 18 | 31.08±1.17 | 0.68 |
| **Motor function and coordination** | | | | | | | | | | | | |
| Rotarod day 1 (latency to fall, s) | 6 | 16 | 56.94±11.97 | 14 | 86.14±13.59 | p=0.12 | 6 | 16 | 141.40±14.97 | 19 | 139.00±12.84 | 0.9 |
| Rotarod day 2 (latency to fall, s) | 6 | 16 | 96.28±15.12 | 14 | 135.40±17.97 | p=0.10 | 6 | 16 | 198.20±16.75 | 19 | 192.80±16.87 | 0.84 |
| Rotarod day 1 (latency to fall, s) | 17 | 16 | 97.69±8.25 | 14 | 88.79±11.71 | p=0.53 | 17 | 16 | 99.38±12.03 | 18 | 122.00±11.14 | 0.18 |
| Rotarod day 2 (latency to fall, s) | 17 | 16 | 145.3±8.56 | 14 | 127.7±11.85 | p=0.23 | 17 | 16 | 143.20±14.43 | 18 | 150.40±12.92 | 0.71 |
| Grip strength (a.u.) | 6 | 16 | 72.42±4.41 | 14 | 74.43±5.00 | 0.76 | 6 | 16 | 71.85±3.55 | 19 | 75.61±3.75 | 0.48 |
| | 13 | 16 | 88.69±3.35 | 14 | 91.12±5.22 | 0.69 | 13 | 16 | 93.98±3.77 | 19 | 84.44±3.96 | 0.09 |
| | 15 | 16 | 88.35±3.83 | 14 | 84.64±5.97 | 0.6 | 15 | 16 | 93.50±4.67 | 19 | 89.53±6.38 | 0.63 |
| Beam balance 10 mm (escape latency, s) | 13 | 16 | 100.00±12.23 | 14 | 113.00±10.08 | 0.36 | 13 | 16 | 100.00±25.86 | 19 | 75.60±14.62 | 0.75 |
| Beam balance 8 mm (escape latency, s) | 13 | 16 | 100.00±8.40 | 14 | 98.25±9.54 | 0.92 | 13 | 16 | 100.00±20.63 | 19 | 90.99±17.93 | 0.86 |
| **Hearing** | | | | | | | | | | | | |
| Acoustic startle at 65 dB (a.u.) | 6 | 16 | 0.87±0.09 | 14 | 0.95±0.15 | 0.6 | 6 | 16 | 0.82±014 | 19 | 0.74±0.10 | 0.73 |
| Acoustic startle at 120 dB (a.u.) | 6 | 16 | 2.88±0.49 | 14 | 3.59±0.56 | 0.34 | 6 | 16 | 2.82±0.32 | 19 | 3.53±0.46 | 0.23 |
| Acoustic startle at 65 dB (a.u.) | 18 | 15 | 0.93±0.07 | 13 | 0.85±0.11 | 0.55 | 20 | 11 | 0.87±0.12 | 12 | 1.00±0.12 | 0.42 |
| Acoustic startle at 120 dB (a.u.) | 18 | 15 | 2.25±0.31 | 13 | 1.87±0.30 | 0.39 | 20 | 11 | 2.33±0.31 | 12 | 2.17±0.25 | 0.7 |
| **Vision** | | | | | | | | | | | | |
| Cued platform training day 1 (escape latency, s) | 19 | 16 | 25.30±3.35 | 14 | 40.59±6.14 | 0.03 | 13 | 16 | 24.46±3.98 | 19 | 28.23±3.93 | 0.61 |
| Cued platform training day 2 (escape latency, s) | 19 | 16 | 6.13±0.76 | 14 | 10.77±2.92 | 0.07 | 13 | 16 | 11.90±2.98 | 19 | 12.08±2.42 | 0.42 |
| **Heat/pain perception** | | | | | | | | | | | | |
| Hot plate (latency to retract, s) | 13 | 16 | 21.40±1.12 | 14 | 22.95±1.46 | 0.69 | 13 | 16 | 17.25±1.00 | 19 | 15.44±0.95 | 0.22 |
| | 15 | 16 | 16.96±1.24 | 14 | 15.81±1.32 | 0.42 | 15 | 16 | 15.89±0.83 | 19 | 15.34±0.60 | 0.59 |

*Table 1 continued on next page*

*Table 1 continued*

| | Males | | | | | | Females | | | | | |
|---|---|---|---|---|---|---|---|---|---|---|---|---|
| | 17 | 16 | 13.12±1.26 | 14 | 12.13±1.02 | 0.71 | 17 | 16 | 16.04±1.00 | 18 | 15.37±1.04 | 0.65 |
| | 20 | 15 | 13.95±0.96 | 12 | 13.71±0.81 | 0.85 | 20 | 11 | 14.26±0.83 | 12 | 12.59±1.30 | 0.3 |
| **Sensorimotor gating** | | | | | | | | | | | | |
| Pre-pulse inhibition (Ø, %) | 6 | 16 | 43.36±7.85 | 14 | 49.84±5.20 | 0.79 | 6 | 16 | 33.53±6.65 | 19 | 40.76±9.97 | 0.13 |
| | 20 | 15 | 16.14±6.88 | 13 | 10.82±5.24 | 0.55 | 20 | 11 | 8.24±3.56 | 12 | 5.62±8.94 | 0.8 |
| **Social behavior** | | | | | | | | | | | | |
| Vocalization (calls, #) | 6 | 16 | 333.3±71.49 | 14 | 231.1±60.22 | 0.29 | n/a | n/a | n/a | n/a | n/a | n/a |
| Vocalization (latency to first call, s) | 6 | 16 | 62.23±18.53 | 14 | 67.14±16.86 | 0.15 | n/a | n/a | n/a | n/a | n/a | n/a |
| Social preference (time spent in pheromone box, %) | n/a | n/a | n/a | n/a | n/a | n/a | 6 | 15 | 64.80±9.63 | 13 | 48.38±6.98 | 0.12 |
| Social preference (time spent in control box, %) | n/a | n/a | n/a | n/a | n/a | n/a | 6 | 15 | 29.80±9.01 | 13 | 36.31±6.02 | 0.21 |
| **Depression-like behavior** | | | | | | | | | | | | |
| Sucrose preference (visits to sucrose corner, %) | n/a | n/a | n/a | n/a | n/a | n/a | 6 | 15 | 44.20±2.48 | 13 | 43.45±3.66 | 0.86 |
| Sucrose preference (visits to water corner, %) | n/a | n/a | n/a | n/a | n/a | n/a | 6 | 15 | 18.19±2.34 | 13 | 20.85±2.34 | 0.36 |
| **Cognitive performance in IntelliCage** | | | | | | | | | | | | |
| Place learning (visits to target corner, %) | n/a | n/a | n/a | n/a | n/a | n/a | 6 | 15 | 35.78±2.33 | 13 | 34.39±2.05 | 0.66 |
| Reversal learning (visits to target corner, %) | n/a | n/a | n/a | n/a | n/a | n/a | 6 | 15 | 37.82±2.79 | 13 | 38.29±2.61 | 0.91 |
| Episodic-like memory acquisition (delta visits to target corner, %) | n/a | n/a | n/a | n/a | n/a | n/a | 6 | 15 | 3.73±2.79 | 13 | 2.31±2.07 | 0.69 |
| Episodic-like memory retrieval (delta visits to target corner, %) | n/a | n/a | n/a | n/a | n/a | n/a | 6 | 15 | 7.29±4.13 | 13 | 3.04±4.40 | 0.49 |
| **Spatial memory and reversal (Morris water maze)** | | | | | | | | | | | | |
| Probe: total distance (cm) | 19 | 16 | 944.5±81.60 | 14 | 1137±57.98 | 0.07 | 13 | 16 | 1258±38.34 | 19 | 1328±31.19 | 0.16 |
| Probe: average velocity (cm/s) | 19 | 16 | 10.50±0.91 | 14 | 12.63±0.64 | 0.07 | 13 | 16 | 13.98±0.43 | 19 | 14.76±0.35 | 0.16 |
| Probe: time in target zone (s) | 19 | 16 | 32.61±3.80 | 14 | 36.39±3.03 | 0.45 | 13 | 16 | 27.03±1.78 | 19 | 26.14±1.38 | 0.69 |
| Probe: visits to target zone (#) | 19 | 16 | 9.44±0.95 | 14 | 11.36±0.58 | 0.11 | 13 | 16 | 14.63±0.84 | 19 | 15.32±0.72 | 0.53 |

*Table 1 continued*

| | Males | | | | | | Females | | | | | |
|---|---|---|---|---|---|---|---|---|---|---|---|---|
| Probe: latency to target zone (s) | 19 | 16 | 8.16±1.71 | 14 | 5.81±1.00 | 0.44 | 13 | 16 | 6.00±1.97 | 19 | 3.74±0.55 | 0.67 |
| Reversal: total distance (cm) | 19 | 16 | 1091±71.27 | 14 | 1236±80.26 | 0.19 | 13 | 16 | 1256±36.02 | 19 | 1442±30.55 | 0.77 |
| Reversal: average velocity (cm/s) | 19 | 16 | 12.12±0.79 | 14 | 13.73±0.89 | 0.19 | 13 | 16 | 13.96±0.40 | 19 | 13.80±0.34 | 0.77 |
| Reversal: time in target zone (s) | 19 | 16 | 28.60±1.81 | 14 | 25.60±2.27 | 0.3 | 13 | 16 | 24.65±1.44 | 19 | 25.39±1.24 | 0.7 |
| Reversal: visits to target zone (#) | 19 | 16 | 8.63±0.62 | 14 | 10.21±0.78 | 0.12 | 13 | 16 | 15.56±0.94 | 19 | 14.42±0.68 | 0.32 |
| Reversal: latency to target zone (s) | 19 | 16 | 7.72±1.41 | 14 | 8.40±1.39 | 0.63 | 13 | 16 | 4.42±0.66 | 19 | 4.39±0.58 | 0.96 |
| **Exploration and working memory** | | | | | | | | | | | | |
| Y maze (arm visits, #) | 20 | 15 | 18.33±1.54 | 13 | 19.77±1.85 | 0.55 | 20 | 11 | 16.09±2.04 | 12 | 19.17±2.32 | 0.33 |
| Y maze (alternation, %) | 20 | 15 | 60.34±3.16 | 13 | 56.85±3.41 | 0.46 | 20 | 11 | 66.53±3.69 | 12 | 60.48±4.52 | 0.32 |
| **Catatonic signs** | | | | | | | | | | | | |
| Bar test (time immobile on the bar, s) | 2 | 15 | 0.35±0.16 | 14 | 0.51±0.28 | p=0.51 | 2 | 16 | 0.23±0.09 | 19 | 0.45±0.21 | 0.74 |
| | 3 | 15 | 0.14±0.03 | 14 | 0.88±0.34 | p=0.19 | 3 | 17 | 0.11±0.01 | 20 | 0.52±0.13 | <0.01 |
| | 5 | 15 | 0.39±0.14 | 14 | 2.63±0.82 | p<0.05 | 5 | 17 | 0.11±0.01 | 19 | 0.27±0.07 | <0.05 |
| | 18 | 16 | 0.54±0.21 | 14 | 1.19±0.29 | p<0.01 | 18 | 10 | 0.10±0.00 | 12 | 0.25±0.09 | <0.05 |
| | 21 | 15 | 0.27±0.11 | 13 | 2.06±0.70 | p<0.01 | 22 | 11 | 0.10±0.00 | 12 | 0.28±0.08 | <0.01 |
| **Executive function and motor skill learning** | | | | | | | | | | | | |
| Hurdle test trial 1 (ratio, s/(crossings +1)) | 6 | 16 | 10.30±1.72 | 14 | 8.34±0.98 | p=0.89 | 6 | 17 | 4.99±0.48 | 19 | 10.80±1.82 | <0.001 |
| Hurdle test trial 2 (ratio, s/(crossings +1)) | 6 | 16 | 1.75±0.17 | 14 | 1.81±0.33 | p=0.52 | 6 | 17 | 1.24±0.17 | 19 | 1.46±0.22 | 0.44 |
| Hurdle test trial 1 (ratio, s/(crossings +1)) | 20 | 16 | 6.76±1.16 | 14 | 11.28±2.04 | p<0.05 | 20 | 11 | 4.88±0.85 | 12 | 9.57±3.14 | 0.08 |
| Hurdle test trial 2 (ratio, s/(crossings +1)) | 20 | 16 | 5.49±1.02 | 14 | 8.07±1.34 | p<0.01 | 20 | 11 | 3.26±0.41 | 12 | 6.99±2.51 | 0.17 |
| Puzzle box (∅ escape latency, s): open | 20 | 16 | 89.48±19.73 | 14 | 118.10±19.82 | p=0.22 | 20 | 11 | 96.23±24.28 | 12 | 140.6±20.26 | 0.17 |
| Puzzle box (∅ escape latency, s): gateway | 20 | 16 | 48.38±8.61 | 14 | 69.95±15.01 | p=0.12 | 20 | 11 | 25.15±6.67 | 12 | 48.18±8.34 | <0.01 |
| Puzzle box (∅ escape latency, s): gateway+bedding | 20 | 16 | 41.96±8.11 | 14 | 71.88±11.72 | p<0.01 | 20 | 11 | 18.21±3.10 | 12 | 53.01±8.19 | <0.0001 |
| Puzzle box (∅ escape latency, s): paper tissue | 20 | 16 | 85.02±10.21 | 14 | 148.30±16.58 | p<0.01 | 20 | 11 | 104.5±32.19 | 12 | 224.3±32.40 | <0.05 |

Table 1 continued

| | Males | | | | | | Females | | | | | |
|---|---|---|---|---|---|---|---|---|---|---|---|---|
| Puzzle box (Ø escape latency, s): nesting material | 20 | 16 | 245.50±21.02 | 14 | 259.70±19.55 | p=0.63 | 20 | 11 | 200.6±35.48 | 12 | 322.3±15.65 | <0.01 |
| Complex running wheel (total distance, m, Ø of 12 hr) | 23 | 8 | 1283.14±319.78 | 8 | 550,23±104.3 | p=0.047 | 23 | 7 | 1956.16±146.65 | 8 | 1118.77±253.23 | 0.04 |
| Complex running wheel (total time, min, Ø of 12 hr) | 23 | 8 | 119.0±29.27 | 8 | 54.78±10.03 | p=0.057 | 23 | 7 | 206.5±15.40 | 8 | 113.4±26.17 | 0.019 |
| Complex running wheel (maximal velocity, m/s, Ø of 12 hr) | 23 | 8 | 0.0555±0.0266 | 8 | 0.0272±0.0103 | p=0.574 | 23 | 7 | 0.0639±0.0226 | 8 | 0.0463±0.0214 | 0.582 |
| Complex running wheel (average velocity, m/s, Ø of 12 hr) | 23 | 8 | 0.0803±0.0311 | 8 | 0.0648±0.0248 | p=0.362 | 23 | 7 | 0.0812±0.0291 | 8 | 0.0686±0.0266 | 0.939 |

2.3 cm higher than in our previous experiments (*Garcia-Agudo et al., 2019*; *Figure 4E*). We measured the time it took forebrain PLP cKO mice and controls to reach the periphery of this arena. Since motor performance was not affected, a longer than normal latency reflects a reduced goal-directed orientation toward the periphery, an executive psychomotor function, perhaps together with dampened motivational force. While behaving normally at age 6 months, when analyzed at the age of 20 months, male PLP cKO mice required more time to reach the periphery in two of two trials (*Figure 4F*). Female PLP cKO mice displayed the longer delay already at the age of 6 months, but when tested again at 20 months only with a non-significant trend (p=0.08). This demonstrates contributory effects of both, sex and age, in this paradigm. Interestingly, in subsequent trials (i.e. comparing time points T1 and T2 for each age group), mutant mice often lost this abnormal delay, which confirms an abnormal executive function rather than an underlying motor impairment.

Finally, we assessed the performance of PLP cKO and control mice at advanced age (23 months) on the 'complex running wheel' (CRW), a paradigm testing cortical motor learning that involves myelinating oligodendrocytes (*Figure 5A*; *McKenzie et al., 2014*). Both female and male groups displayed reduced total running distance and times (male total time analysis p=0.056) (*Figure 5B and C*). Since the maximal and average running velocities did not differ between cKO and control groups (*Figure 5D and E*), this performance difference likely reflects a reduced drive rather than altered motor learning. This would be in accordance with the reduced executive function in the puzzle box and hurdle test.

## Discussion

We report the generation and behavioral analysis of a novel mouse mutant, in which the lack of oligodendroglial *Plp1* expression was spatially restricted to neocortex, hippocampus, and corpus callosum by targeting *Plp1* in ventricular zone stem cells of the embryonic forebrain by usage of *Emx^{IRESCre}*. This allowed behavioral tests of higher brain functions that rely on motor output, because the myelinating oligodendrocytes in cerebellum and spinal cord, which contribute to basic motor-sensory functions, were spared.

Forebrain glutamatergic neurons are also derived from the Emx lineage (*Gorski et al., 2002*) and are therefore *Plp1* mutant. However, the myelin-specific protein PLP is not required for neuronal functions, as directly shown by deleting *Plp1* in excitatory forebrain neurons with the help of Nex-Cre mice (*Lüders et al., 2017*). All *Plp1* KO-linked phenotypes, including axonal pathology and neuroinflammation, are caused by PLP deletion in oligodendrocytes and not neurons (*Lüders et al., 2017*). Thus, also phenotypic effects of forebrain-specific PLP cKO mice are caused by PLP-deficient oligodendrocytes.

The *Plp1* gene was chosen as a target, because the cardinal features of PLP deficiency are slowly progressive axonopathy in the presence of close to normal amounts of myelin, with ultrastructural features reminiscent of advanced brain aging (*Griffiths et al., 1998*; *Janova et al., 2018*). Moreover,

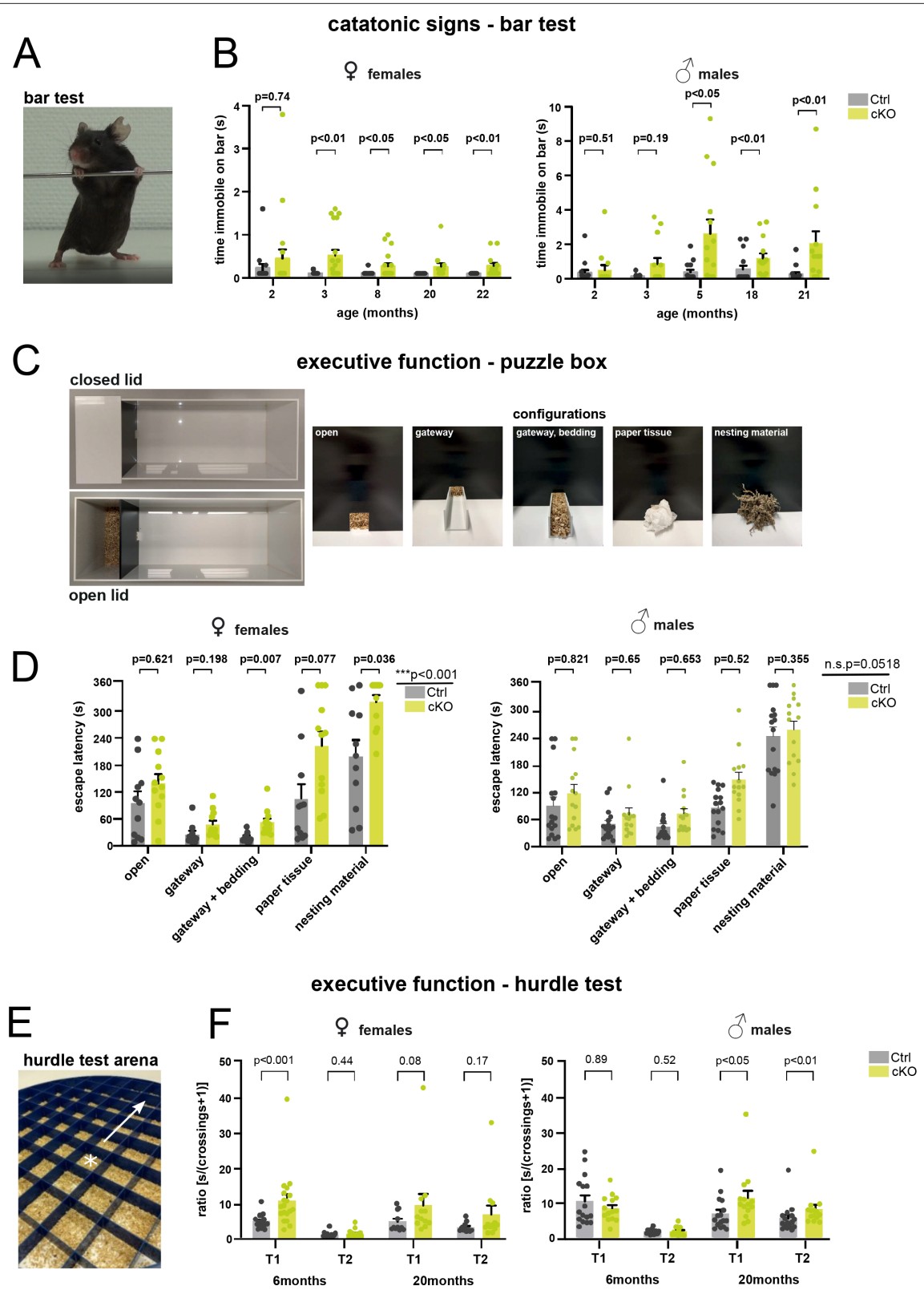

**Figure 4.** Forebrain proteolipid protein (PLP) conditional knockout (cKO) mice display catatonia-like signs along with impaired executive function. (**A**) Representative image of a mouse subjected to the bar test. The time a mouse remains immobile upon placement of its forelimbs on a horizontal bar is measured. (**B**) Bar test of female and male groups of forebrain PLP cKO (n=12-20 or 13-14, respectively) and Ctrl (n=10-17 or 15-16, respectively) mice were tested at the indicated ages. Bars represent means ± SEM; datapoints represent individual mice. Mann-Whitney U-tests wereperformed for

*Figure 4 continued on next page*

*Figure 4 continued*

statistical analyses. (**C**) Puzzle box setup with closed and open lid. The time to escape from a well-lit open space into a dark shelter with increasingly difficult configurations (open entry, addition of a gateway, gateway filled with bedding, entry filled with paper tissue, entry filled with nesting material) is taken as a readout. (**D**) Female and male groups of forebrain PLP cKO (n=12 or 14, respectively) and Ctrl (n=11 or 16, respectively) mice were assessed at age 20 or 18 months, respectively. Bars represent means ± SEM; datapoints represent individual mice. Repeated measures two-way analysis of variance (ANOVA); significance level for the factor 'genotype' is given on the left; Sidak multiple comparison test was conducted as post hoc test and p-values are given in the graph. (**E**) Modified hurdle test assessing the escape latency of mice by crossing 5 cm hurdles toward the periphery (arrow) upon placement into the center (marked by a star) of a circular open field arena. (**F**) Female and male groups of forebrain PLP cKO (n=11-12 or 14, respectively) and Ctrl (n=17-19 or 16, respectively) mice were assessed at the age of 6 and 20 months. trial 1 or 2, T1 or T2. Bars represent mean ± SEM; datapoints represent individual mice. Mann-Whitney U-test was performed for statistical analysis. Levels of significance were defined as p>0.05 (*), p>0.01 (**) and p>0.001 (***). The precise number of mice tested per experiment, mean, SEM, and p-values are specified in *Table 1*.

the *Plp1* null mutation in mice is a genetic model of human SPG2 (*Garbern et al., 2002*). Conventional *Plp1* null mice (*Klugmann et al., 1997*) have been recently studied on a behavioral scope (*Gould et al., 2018*), and these results can be directly compared with the results obtained in our new forebrain PLP cKO mouse model.

*Plp1* null myelin has been extensively studied in the past and subtle ultrastructural abnormalities have been described (*Klugmann et al., 1997*; *Möbius et al., 2016*; *Patzig et al., 2016*; *Rosenbluth et al., 2006*; *Steyer et al., 2023*). We have shown that PLP, which provides structural stability to compact myelin, also stabilizes the architecture of the 'myelinic channel system' (*Steyer et al., 2020*). The latter is a cytosolic space that connects the oligodendroglial cell body with the peri-axonal myelin compartment that contains transporters involved in axonal metabolic support. We could further show that conventional *Plp1* KO mice, which are well myelinated, exhibit lower ATP levels in axons of the optic nerves compared to controls, when analyzed with transgenically expressed metabolic sensors (*Trevisiol et al., 2020*). This makes it likely that conduction blocks, axonal swellings, and neurodegeneration are caused by perturbed axonal energy metabolism (*Fünfschilling et al., 2012*; *Lee et al., 2012*; *Saab et al., 2016*; *Trevisiol et al., 2017*). In addition, axonal degeneration leads to secondary neuroinflammation. We note that other myelin mutants, such as *Cnp* KO mice (*Lappe-Siefke et al., 2003*), exhibit an even earlier onset of axonal pathology that virtually coincides with the onset of gliosis (*Edgar and Nave, 2009*). This raises the possibility that inflammation in myelinated tracts is a more general response of resident microglia to oligodendrocyte dysfunctions, possibly reflecting the failure to deliver lactate to the axonal compartment and its release into the extracellular space. It is this secondary neuroinflammation that contributes to reversible catatonic signs in myelin mutant mice (*Janova et al., 2018*) and that can now be defined as a clinical feature of forebrain-specific white matter disturbance.

Theoretically, the specific behavioral defects of *Plp1* cKO mice could be caused by (1) the moderate decrease of the axonal conduction velocity that was previously reported (*Gould et al., 2018*; *Gutiérrez et al., 1995*), (2) the progressive axonopathy that is caused by energy deficits and is likely preceded by conduction blocks in fiber tracts that appear morphologically intact (*Trevisiol et al., 2020*), (3) the secondary gliosis, as also documented for *Cnp* KO mice (*Garcia-Agudo et al., 2019*; *Janova et al., 2018*), another mutant with structural abnormalities of myelin (*Lappe-Siefke et al., 2003*; *Snaidero et al., 2017*), or (4) any combination hereof. The age-dependent increase of symptom severity suggests that the responsible mechanisms include progressive myelin-dependent axonopathy (with conduction blocks) plus neuroinflammation rather than subtle conduction delays. That is in agreement with the 'rescue' of the catatonic signs, which we consider a readout of reduced executive functions, in Cnp mutant mice by microglial depletion (*Janova et al., 2018*). We also note that reduced conduction velocity by itself has little impact on cortical processing, as we have shown for auditory signals in *shiverer* mice (*Moore et al., 2019*).

With respect to CNS regions, the present work further uncouples the role of myelinated tracts for basic motor performance and executive functions. Conventional PLP KO mice exhibit first behavioral abnormalities at the age of 3 months (*Gould et al., 2018*), that is, before the impairment of motor performance that we had initially determined by rotarod experiments (*Griffiths et al., 1998*). Here, we show in PLP cKO mice that the forebrain-specific loss of axon-myelin integrity impairs selectively executive functions that require the multimodal integration of cortical processing upstream of any motor output. Motor performance itself was spared at any age tested. Also olfactory and thalamic input as well as emotionally relevant input into (and from) the amygdala should be less affected in

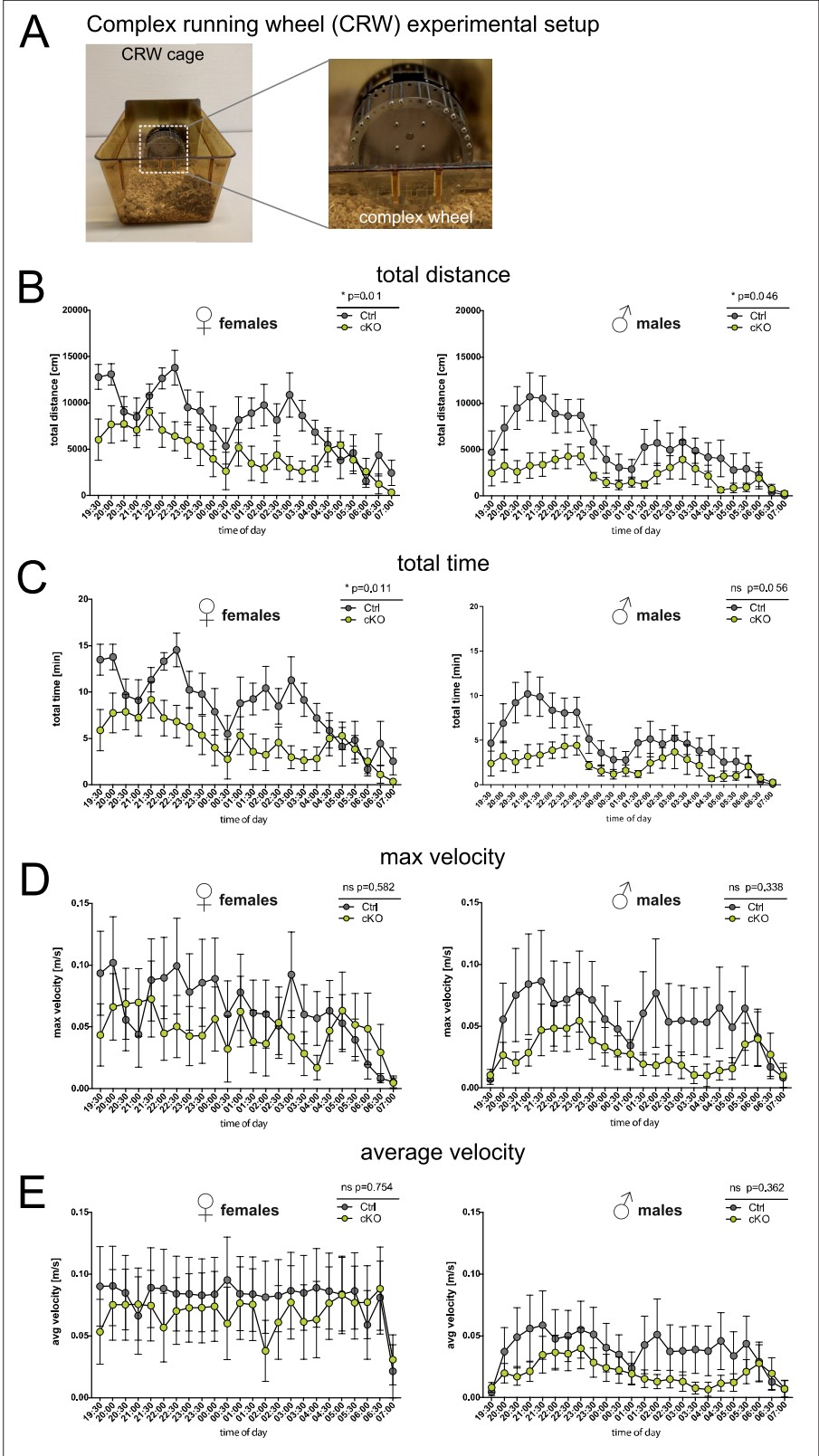

**Figure 5.** Forebrain proteolipid protein (PLP) conditional knockout (cKO) mice display reduced running distance but normal velocity in the complex running wheel (CRW) paradigm. (**A**) Experimental setup to study voluntary running behavior and motor skill learning in the CRW. (**B**) Total running distance in the CRW of female and male groups of PLP cKO and Ctrl mice at age 23 months. (**C**) Total running time of female and male groups of PLP cKO

*Figure 5 continued on next page*

*Figure 5 continued*

and Ctrl mice at age 23 months in the CRW. (**D**) Maximal velocity reached of female and male groups of PLP cKO and Ctrl mice at age 23 months in the CRW. (**E**) Average velocity of female and male groups of PLP cKO and Ctrl mice at age 23 months in the CRW. Both female and male cKO mice displayed reduced running distance but normal average running velocity. A trend toward reduced running time reached significance in female but not in male cKO mice. Note that the result is interpreted as reflecting that cKO mice show normal motor capability but impaired executive function. A total of 7-8 mice per sex and genotype were used. Superimposed plot with connecting lines; repeated measures two-way analysis of variance (ANOVA); not significant, n.s.; *, p<0.05.

forebrain PLP cKOs, which could explain different behaviors of conventional PLP KO mice (*Gould et al., 2018*). The principle finding that PLP-dependent processes affecting myelin and axonal integrity in the forebrain cause the impairment of executive functions is relevant to higher brain functions known to be affected across psychiatric diseases.

# Materials and methods
## Mouse model
To delete the *Plp1* gene in the forebrain we interbred *Plp^{flox}* mice in which exon 3 of the *Plp1* gene is flanked by loxP sites (*Lüders et al., 2019*; *Lüders et al., 2017*; *Wang et al., 2017*) with mice expressing Cre recombinase under control of the *Emx1* gene promoter (*Gorski et al., 2002*) on C57Bl/6N background. Genotyping was as reported previously (*Gorski et al., 2002*; *Lüders et al., 2017*). Experimental male *Plp^{flox/Y}*Emx^{IREScre}* and female *Plp^{flox/flox}*Emx^{IREScre}* mice are termed cKO whereas male *Plp^{flox/Y}* and female *Plp^{flox/flox}* mice served as controls (Ctrl). Mice were bred and kept in the animal facility of the Max Planck Institute of Experimental Medicine with a 12 hr light/dark cycle and two to five mice per cage. All experiments were performed in accordance with the German Animal Protection Law (TierSchG) and approved by the Niedersächsisches Landesamt für Verbraucherschutz und Lebensmittelsicherheit (LAVES); License numbers were 33.19-42502-04-15/1833 and 33.19-42502-04-18/2803.

## Light-sheet microscopy
LSM to detect PLP labeling displayed in *Figure 1A and B* was performed as follows. Animals were sacrificed using $CO_2$ followed by transcardial perfusion with HBSS and paraformaldehyde (PFA) (4%) in PBS. Brains were extracted and postfixated in 4% PFA/PBS overnight and stored in PBS at 4°C until further use. Brains were cut using a 1 mm custom-made brain matrix and subjected to whole-mount staining and clearing. Samples were dehydrated in a methanol/PBS series (50%, 80%, 100%) followed by overnight bleaching and permeabilization in a mixture of 5% $H_2O_2$/20% dimethyl sulfoxide (DMSO) in methanol at 4°C. Samples washed further in methanol prior to incubation in 20% DMSO in methanol at RT for 2 hr. Samples were then rehydrated using a descending methanol/PBS series (80%, 50%, PBS) and further washed with in PBS/0.2% Triton X-100 for 2 hr. The samples were then incubated overnight in 0.2% Triton X-100, 20% DMSO, and 0.3 M glycine in PBS at 37°C and blocked using PBS containing 6% goat serum, 10% DMSO, and 0.2% Triton X-100 for 2 days at 37°C. Samples were retrieved, washed twice in PBS containing 0.2% Tween 20, and 10 µg/ml heparin (PTwH) at RT for 1 hr and incubated with primary antibody solution (rat anti-PLP [aa3, 1:250], mouse anti-CNP [Atlas, 1:250]) for 14 days at 37°C. After several washes in PTwH, samples were incubated with secondary antibody solution (goat anti-rat Alexa 555; goat anti-mouse Alexa 633) for 7 days at 37°C. Prior to clearing, the samples were again washed in PTwH and brain slices were embedded in 2% Phytagel in PBS. Slices were dehydrated using an ascending series of methanol/PBS (20%, 40%, 60%, 80%, 2×100% 1 hr, RT) followed by overnight incubation in a mixture of 33% dichloromethane (DCM) and 66% methanol at RT. Samples were further delipidated by incubation in 100% DCM for 40 min and transferred to pure ethyl cinnamate (Eci; Sigma-Aldrich #112372) as clearing agent.

LSM was performed using a LaVision Ultramicroscope II equipped with a 2× objective, corrected dipping cap, and zoom body. Slices embedded in Phytagel tubes were mounted onto the sample holder. Images were acquired in mosaic acquisition mode with the following specifications: 5 µm sheet thickness; 30% sheet width; 2× zoom; 2×3 tiling; 4 µm z-step size; dual site sheet illumination; 100 ms

camera exposure time. Green, red, and far-red fluorescence were recorded using 488 nm, 561 nm, and 633 laser excitation (30%) and respective emission filters.

Images were loaded into Vision4D 3.3 (Arivis) and stitched using the tile sorter setup. Datasets were pseudocolored and visualized in 3D (maximum intensity mode).

## Immunohistochemistry and immunofluorescence stainings

Immunohistochemistry for neuropathological analysis displayed in *Figure 2* and *Figure 2—figure supplement 1* was performed as previously described (*Lüders et al., 2019*; *Patzig et al., 2016*). Antibodies were specific to MAC3 (1:400; Pharmingen 553322; clone M3/84), IBA1 (1:1000; Wako 019-19741), APP (1:1000; Millipore MAB 348) or GFAP (1:200; Novocastra NCL-GFAP-GA5). Images were captured at 20× (GFAP, IBA1, MAC3) or 40× (APP) magnification using a bright-field light microscope (Zeiss AxioImager Z1) coupled to a Zeiss AxioCam MRc camera controlled by Zeiss ZEN 1.0 software and processed using Fiji.

For immunofluorescence staining, antibodies specific to PLP (rat anti-PLP [aa3, 1:250]), MBP (1:1000; custom Klaus-Armin Nave/Kathrin Kusch), GFAP (anti-GFAP [mouse, GA5, Leica, 1:200], and IBA1 [1:1000 Wako]) were used and visualized with appropriate secondary antibodies (anti rat Dylight650, anti rabbit Dylight 650, anti mouse Dylight 555; Thermo Fisher Scientific). Epifluorescence microscopy was performed on a Zeiss Observer Z1 microscope equipped with Plan-Apochromat 20×/08 and Fluar 2.5×/0.12 objectives, a Colibri 5 LED light source (630 nm, 555 nm, 475 nm, 385 nm excitation), 96 HE BFP, 90 HE DAPI/GFP/Cy3/Cy5, 38 GFP, 43 DsRed, 50 Cy5 Zeiss filter sets, a Axiocam MrM, and an SMC900 motorized stage. For whole-brain slice microscopy, a preview scan at 2.5× magnification was taken and focus support points were distributed and manually set for imaging at 20× magnification in the ZEN imaging software (Zeiss).

To quantify axonal swellings, the region of interest was selected and APP-immunopositive axonal swellings were counted. To quantify brain region immunopositive for IBA1, MAC3, or GFAP, the region of interest was selected and analyzed using an ImageJ plug-in (*de Monasterio-Schrader et al., 2013*; *Lüders et al., 2019*; *Lüders et al., 2017*) for semi-automated analysis. Per genotype (Ctrl, cKO), five to seven mice were assessed at the age of 22 months as indicated by the datapoints in *Figure 2* and *Figure 2—figure supplement 1*, in which n-numbers represent individual mice. Assessment of neuropathology was performed blinded to the genotype. For quantification of immunofluorescence stainings, the respective region of interest was demarcated and the staining positive area was determined by automated thresholding in ImageJ.

## Immunoblotting

Prefrontal cortex (frontal to bregma 5.22) and cerebellum were dissected from mice and homogenized in 1× TBS with protease inhibitor (Complete Mini, Roche). Protein concentration was measured using the DC protein assay (Bio-Rad). Immunoblotting displayed in *Figure 1C and D* was essentially as described (*Kusch et al., 2017*; *Schardt et al., 2009*). Briefly, lysates from prefrontal cortex (3.2 µg for PLP/DM20 and 1 µg for actin) and cerebellum (0.8 µg for PLP/DM20 and 1 µg for actin) were separated on 15% SDS-polyacrylamide gels and blotted onto PVDF membranes (Hybond, Amersham) using the Novex Semi-Dry Blotter (Invitrogen). Primary antibodies were incubated in 5% milk powder in TBST over night at 4°C. Primary antibodies were specific for PLP/DM20 (A431; 1:5000; *Jung et al., 1996*), MBP (1:500; DAKO), and actin (1:1000 for cerebellum, 1:5000 for prefrontal cortex; Sigma). Secondary HRP-coupled antibodies (dianova) were detected using the ChemoCam system (Intas).

## Mouse maintenance and behavioral tests

All mice for behavioral tests were maintained in ventilated cabinets (Scantainers, Scanbur, Karlslunde, Denmark), separated by gender, under standard laboratory conditions, including a 12 hr light/dark cycle (lights off at 7 PM) at 20–22°C, 50–60% humidity, and with access to food and water ad libitum. Upon weaning, mice were separated by gender and genotype and remained group-housed in standard plastic cages (two to five mice per cage). Male mice were single-housed at age of 6 months, due to requirements of experimental tests. Male and female cKO mice (N=14 and 20, respectively) were compared to respective control mice (N=16 and 17, respectively) in all behavioral experiments. A series of behavioral paradigms, described below, was conducted on mice throughout lifespan, covering ages between 2 and 23 months (*Table 1*). General health status of mice was continuously

monitored by body weight, home-cage observation including general activity and appearance, nest building and interaction with littermates. Group sizes decreased upon progressing age due to natural death. In all behavioral experiments, mice were allowed to habituate to conditions of the experimental room for a minimum of 30 min prior to testing. Male and female mice were tested separately.

## Muscle strength, motor coordination, and motor learning
### Rotarod – motor coordination and learning
The rotarod (ENV-577M, Med Associates Inc Georgia, VT, USA) was performed as previously described (*Dere et al., 2014*). Briefly, all mice were tested in a total of two trials over 2 consecutive days. Each trial consisted of a maximum of 5 min, in which mice had to run continuously on a horizontal rotating drum accelerating from 4 to 40 rpm. The latency to fall (s) was assessed for each mouse during both trials. Mice of both genders were tested in the rotarod at the age of 6 and 17 months.

### Grip strength – forelimb muscle strength
The forelimb grip strength of male and female mice was assessed at age 6, 13, and 15 months. The test was performed as previously described (*Netrakanti et al., 2015*). Briefly, each mouse was tested in a total of three consecutive trials. During each trial, mice were lifted gently by their tails and allowed to grasp a wire of the grip strength meter (TSE Systems, Bad Homburg, Germany). Upon grasping the wire, mice were brought into a horizontal position before being gently pulled back by their tails, to assess the applied forelimb force. The average grip strength per mouse was calculated.

### Beam balance – motor coordination
Motor coordination was assessed at age 13 months, using beam balance test as previously described (*Netrakanti et al., 2015*). Briefly, the ability of mice to cross elevated beams (59 cm length) of decreasing diameter (25, 10, or 8 mm) was measured. Mice were placed at the illuminated end of the respective beam and the latency to reach the target zone, an attached cardboard cage with bedding, was recorded. The test was conducted over 3 consecutive days, starting with the 25 mm beam (day 1), then 10 mm (day 2) and 8 mm beam (day 3). Each mouse was tested in a maximum of three trials per day, in case it fell off the beam, and given 60 s to reach the target.

## Hearing and sensorimotor gating
### Pre-pulse inhibition of the acoustic startle response
Male and female mice were tested for both general hearing and sensorimotor gating at age of 6 and 18 or 6 and 20 months, respectively. A detailed protocol of pre-pulse inhibition (PPI) was reported previously (*Netrakanti et al., 2015*). Sensorimotor gating experiments were conducted within sound attenuating chambers (TSE Systems, Bad Homburg, Germany), in which acoustic stimuli (120 dB) both with and without a preceding pre-pulse evoked startle responses, recorded by a force-sensitive platform. PPI, using a 70, 75, or 80 dB pre-pulse, was calculated by the following formula: %PPI = 100 – [(startle amplitude after pre-pulse)/(startle amplitude after pulse only) × 100] (*Pan et al., 2019*). To assess hearing abilities of male and female mice, the amplitude of response (arbitrary units) to 65 dB background noise as well as to six pulse-alone trials with startle stimuli of 120 dB was measured.

## Sensory functions
### Hot plate test – nociception
Pain perception in male and female was assessed at the age of 13, 15, 17, and 20 months, using the hot plate test as described (*Dere et al., 2014*). Briefly, mice were placed on a preheated (55°C) metal plate (Ugo Basile Srl, Comerio, Italy) and the latency (s) to retract by jumping or licking of the hind paws was recorded. Mice were exposed for a maximum of 40 s, used as cut-off time, in case they did not show an aversive response to the heated plate.

### Cued platform training in the MWM – vision
General vision of male and female mice was evaluated within the MWM experiment at age 19 or 13 months, respectively, as detailed before (*Dere et al., 2014*). Vision was assessed during the first 2 days of acquisition training, in which an escape platform was submerged 1 cm below the surface of

opaque water. A small blue flag protruding above the water surface was attached at the center of the platform and used as cue for locating the escape platform. Each mouse was tested on 2 consecutive days in four trials per day with a 5 min inter-trial-interval (ITI). Escape latency, velocity of swimming, and path length were recorded with video-tracking system (Viewer3, Biobserve GmbH, Bonn, Germany).

## Social behavior and social preference
### Ultrasound vocalization – communication
Ultrasonic vocalization was evaluated as described in male mice only, at age 6 months (*Dere et al., 2014*; *Hammerschmidt et al., 2012*). Male mice of both genotypes were single-housed 1 week prior to testing. Each male mouse was exposed to an anesthetized unfamiliar C57Bl/6N WT female within its home-cage for 3 min. Number of ultrasonic calls of the resident males and latency to first call (s) were recorded.

### Social boxes in the IntelliCage design – pheromone-based social preference
Pheromone-based social preference was evaluated at age 6 months in female mice only as described (*Dere et al., 2014*). Briefly, mice underwent anesthesia for subcutaneous implantation of ISO standard transponders (PM162-8) below the skin of the neck, 1 day prior to group-housing within IntelliCages. Mice remained in the cages for a total of 6 days for an IntelliCage-based behavioral phenotyping (see below). Social preference was tested on the last day by connecting two social boxes to the left and right side of the cages. Connection of the boxes was carried out via plastic tubes equipped with two-ring RFID antennas, which monitored entrance and exit of individual mice to each box. Initially, both boxes were filled with fresh bedding and animals were allowed to freely explore for a 1 hr habituation. Subsequently, used bedding containing pheromones of male C3H mice was added to one of the boxes and the time spent in the pheromone vs. non-pheromone box was recorded for another hour.

## Cognitive flexibility, episodic-like memory, and anhedonia
### IntelliCage-based behavioral phenotyping battery
By application of our IntelliCage-based behavioral phenotyping design, we assessed multiple facets of cognition as well as sucrose preference, as measure of anhedonia, in female mice only, at age 6 months as described (*Dere et al., 2014*). One day after subcutaneous implantation of transponders (see above), mice were group-housed, separated by genotype, and remained in the IntelliCages for 6 days. Place learning was acquired within the first 24 hr (day 1), in which an individual mouse learned that only one out of four corners was rewarded with water, while the other corners remained blocked. The number of mice assigned to each corner was balanced and semi-randomly determined. On day 2, reversal learning was assessed to measure cognitive flexibility as well as perseveration. Mice had to learn that the previously rewarded corner was now blocked, and instead the diametrically opposed corner now rewarded. Sucrose preference was assessed on day 3 by comparing preference for a corner rewarded with a 2% sucrose solution over another corner rewarded with tap water. During these 24 hr, the two previously blocked corners were now rewarded with either sucrose solution or tap water, whereas the remaining corners were blocked. The visits to the respective target corners during place and reversal learning as well as sucrose preference testing were used for statistical analysis. On days 4–5 mice had again access to two rewarding corners providing either a sucrose solution or tap water. However, these corners were now again the diametrically opposed corners to day 3 and access to the corners was only provided for a limited time, namely during the first 2 hr of the active phase of the mice (6–8 PM). Hence, mice were required to form a multimodal association containing the information on the type of reward provided (*what*) as well as their locations (*where*) and the time at which to expect the reward (*when*), rendering this approach an experimental model for the assessment of episodic memory comparably to humans. Visits to rewarded corners during acquisition (day 4) and retrieval (day 5) of episodic-like memory were recorded and the delta between visits to the corner providing sucrose solution and the corner providing tap water was calculated.

## Spatial memory and learning

### Morris water maze

Spatial memory as well as cognitive flexibility and perseveration, via reversal learning, were evaluated in male and female mice at the age of 19 or 13 months, respectively, using the MWM task. The test was conducted as described (*Dere et al., 2014*; *Netrakanti et al., 2015*). Mice were tested within a circular tank (diameter 1.2 m and depth 0.6 m) filled with opaque water (25±1°C) in various successive phases, starting with 2 days of cued platform training as described above (see: vision). Subsequently, the blue flag cue was detached from the escape platform (10 cm × 10 cm), which was submerged 1 cm below the water surface and relocated within the tank, and mice were tested for 8 days during hidden platform training. Throughout these days mice had to form a spatial memory for the escape platform using various extra-maze cues placed on the walls of the testing room. During all training phases mice were tested in four daily trials with an ITI of 5 min and the individual performances were recorded with a video-tracking system (Viewer3, Biobserve GmbH, Bonn, Germany) for subsequent analysis. Training was followed by a probe test to assess spatial memory and the time spent in the target quadrant of the maze as well as visits and the latency to the target quadrant were used for statistical analysis. Moreover, total distance swam and the average swimming velocity were analyzed, to exclude potential motor deficits. Finally, mice were exposed to 4 days of reversal training, in which the only difference to the hidden platform training was that the escape platform was relocated into a new quadrant of the tank. Reversal training was followed by another probe trial, wherein the same parameters were determined as in the first probe test.

## Working memory

### Y maze continuous alternation

Spontaneous alternation was assessed in male and female mice of 20 months age, in a Y-shaped maze (*Dere et al., 2014*). Mice were individually placed into the center of a triangle-shaped maze with three identical open arms (7.5 cm × 18 cm × 23.5 cm) and allowed to freely explore the maze for a total of 5 min. Performance of all mice including total number of arm entries, defined as entering with all four paws, was recorded with a video-tracking system (Viewer3, Biobserve GmbH, Bonn, Germany). An alternation ratio, defined as the number of alternating triplets multiplied by 100 and divided by the total number of entries, was calculated.

## Catatonic signs and executive function

### Bar test

Catatonic signs were assessed at various ages ranging from 2 to 21 months in male and 2 to 22 months in female mice. The bar test was conducted as described before (*Garcia-Agudo et al., 2019*; *Janova et al., 2018*). Briefly, mice were gently carried by the tail and brought into proximity of a horizontal bar made of stainless steel. Upon grasping the bar with both forepaws and standing upright, the tail was released. Mice were tested in two consecutive trials, which were recorded with a high-resolution camcorder (Sony HDR-CX405, Sony, Tokyo, Japan) and the time spent immobile at the bar (s) was determined.

### Modified hurdle test

The hurdle test is a novel tool to measure executive dysfunction in catatonia-like syndromes (*Garcia-Agudo et al., 2019*). The test setup comprises a circular open field arena containing a polyvinylchloride comb inset (119 cm diameter), made of equally built (10 cm × 10 cm) connected combs, and 140 lux light intensity at the center of the arena, to motivate mice to move to the periphery. We applied minor modifications to the previously reported setup, by increasing the height of the connected combs from 2.7 cm to now 5.0 cm and adding fresh woodchip bedding into each comb. These modifications were applied, to increase the challenge, while simultaneously reducing aversive responses to the novel environment upon introduction of bedding as a familiar and comfortable texture. The experiment was conducted in male and female mice at age 6 and 20 months, respectively, as described (*Garcia-Agudo et al., 2019*). Briefly, mice were placed into the center of the above-mentioned inset and their performance until reaching the periphery or the cut-off time of 5 min, was recorded using the Viewer 3 tracking software (Biobserve GmbH, Bonn, Germany). All mice were tested in two consecutive trials

with a 5 min ITI. Executive performance was assessed by calculating the ratio of latency to periphery (s) divided by the number of crossed hurdles (#). To account for animals that did not overcome any hurdle, we calculated the ratio as [(s)/(#+1)].

## Puzzle box

As an additional test assessing executive function, the puzzle box was conducted on male and female mice at age 20 months, respectively. We employed the test as described (*O'Connor et al., 2014*) with minor modifications. The experimental setup comprises a rectangular-shaped arena (75 cm × 28 cm × 25 cm) split into an enclosed shelter (15 cm × 28 cm) and an illuminated (140 lux) open compartment (60 cm × 28 cm). The compartments are connected to each other via a small doorway, centered at the front wall of the shelter (4 cm width), through which mice can escape into the shelter upon placement into the open compartment. Over a course of 5 consecutive days, mice were required to overcome a total of five challenges of increasing difficulty, within a limited amount of time, to reach the shelter. We employed the following challenges: (1) open doorway, (2) gateway within doorway, (3) gateway filled with bedding, (4) plug made of paper tissue, (5) plug made of nesting material (shredded cardboard paper). Each mouse was tested in a total of three trials per day, in which mice were exposed to two different challenges daily. During the first trial on each day, mice were exposed to the challenge they had to overcome last on the day before, whereas the following two trials measured escape latencies upon introduction of a novel unfamiliar challenge. Exceptions to this approach were the very first trial on day 1 (open entry), which was tested only this one time, and day 5, since mice were tested only once with the most difficult challenge (plug made of nesting material) on day 5. Cut-off time was increased from 4 to 6 min during challenges 4 and 5, to provide sufficient time for mice to be able to unplug the doorway into the shelter. Performance of each mouse was recorded with a video-tracking system (Viewer3, Biobserve GmbH, Bonn, Germany) and averaged escape latency for each challenge measured.

## Complex wheel running

Overnight voluntary CRW was conducted at age 23 months, as an additional measure of both drive and motor-cognitive performance. Mice of both genotypes and genders were single-housed and exposed to CRW (TSE Systems, Bad Homburg, Germany) for 24 hr. Placement of mice into the respective CRW cages was carried out in the morning (8–9 AM) allowing the mice to familiarize themselves with the novel environment throughout the day. CRW are defined by randomly omitted bars, providing a motor-cognitive and coordinatory challenge. Voluntary running on CRW was recorded automatically (Phenomaster software, TSE Systems, Bad Homburg, Germany) yielding information on time spent running (min), total distance run (cm), and running velocity (cm/min). Analysis of CRW performance was conducted over 12 hr during the active phase (lights off from 7 PM to 7 AM).

## Statistical analyses

Computation of an appropriate sample size for this study was carried out via G*Power software (*Faul et al., 2007*) based on the following statistical requirements and assumptions: $\alpha$-error 0.05, $\beta$-error 0.085, statistical effect size 1.000, based on previous IntelliCage-based experiments in our lab. We chose the IntelliCage-based assay serving as a crucial limiting experiment, due to the high cognitive demand of the setup (see also: *Dere et al., 2018*). Mice of both genders (male/female) and both genotypes (control/conditional mutant) were used, with each mouse being a biological replicate in each assay.

Statistical analysis was performed using GraphPad Prism 9.0. Between-group comparisons were made by either one-way or two-way analysis of variance (ANOVA) with repeated measures or t-test for independent samples. Mann-Whitney U, Wilcoxon tests were used if the normality assumption was violated (as assessed by the Kolmogorov-Smirnov test). Data presented in the figures and text are expressed as mean ± SEM; p-values <0.05 were considered significant. Statistical outliers were assessed using the GraphPad Grubbs outlier tool, but no such outliers had to be removed from the data. Only mice of any gender or genotype that did not perform in an experiment were excluded from statistical analyses.

## Acknowledgements

We thank A Fahrenholz and R Jung for technical support, E-M Krämer-Albers for antibodies and the International Max Planck Research School for Genome Science (IMPRS-GS) for supporting SBS. Our work is supported by grants of the Deutsche Forschungsgemeinschaft (WE 2720/4-1, and WE 2720/5-1 to HBW; TRR274 to KAN and HE), the Adelson Medical Research Foundation (AMRF to KAN), and the European Research Council (ERC Advanced grant 'MyeliNano' to KAN).

## Additional information

### Competing interests

Klaus-Armin Nave: Reviewing editor, eLife. The other authors declare that no competing interests exist.

### Funding

| Funder | Grant reference number | Author |
| --- | --- | --- |
| Deutsche Forschungsgemeinschaft | WE 2720/4-1 | Hauke B Werner |
| Deutsche Forschungsgemeinschaft | WE 2720/5-1 | Hauke B Werner |
| Deutsche Forschungsgemeinschaft | TRR274 | Hannelore Ehrenreich |
| Dr. Miriam and Sheldon G. Adelson Medical Research Foundation | | Klaus-Armin Nave |
| European Research Council | ERC Advanced Grant 'MyeliNANO' 671048 | Klaus-Armin Nave |
| European Research Council | 2016-2021 | Klaus-Armin Nave |
| Boehringer Ingelheim Fonds | PhD Fellowship | Constanze Depp |
| Max Planck Institute for Multidisciplinary Sciences | open access funding | Sahab Arinrad |

The funders had no role in study design, data collection and interpretation, or the decision to submit the work for publication.

### Author contributions

Sahab Arinrad, Data curation, Formal analysis, Validation, Investigation, Visualization, Project administration, Writing – review and editing; Constanze Depp, Data curation, Formal analysis, Investigation, Visualization, Writing – review and editing; Sophie B Siems, Maria A Eichel, Data curation, Investigation, Visualization; Andrew Octavian Sasmita, Kurt Hammerschmidt, Investigation, Methodology; Anja Ronnenberg, Investigation; Katja A Lüders, Data curation, Investigation; Hauke B Werner, Data curation, Supervision, Writing – original draft; Hannelore Ehrenreich, Conceptualization, Data curation, Supervision, Funding acquisition, Validation, Project administration, Writing – review and editing; Klaus-Armin Nave, Conceptualization, Resources, Funding acquisition, Project administration, Writing – review and editing

### Author ORCIDs

Sahab Arinrad ⓘD http://orcid.org/0000-0002-7083-1358
Constanze Depp ⓘD http://orcid.org/0000-0003-2868-6932
Sophie B Siems ⓘD http://orcid.org/0000-0002-7760-2507
Andrew Octavian Sasmita ⓘD http://orcid.org/0000-0001-7379-6749
Maria A Eichel ⓘD http://orcid.org/0000-0002-9925-7249
Hauke B Werner ⓘD http://orcid.org/0000-0002-7710-5738

Hannelore Ehrenreich [ID] http://orcid.org/0000-0001-8371-5711
Klaus-Armin Nave [ID] http://orcid.org/0000-0001-8724-9666

### Ethics

All experiments were performed in accordance with the German animal protection law (TierSchG) and approved by the Niedersächsisches Landesamt für Verbraucherschutz und Lebensmittelsicherheit (LAVES) License numbers were 33.19-42502-04-15/1833 and 33.19-42502-04-18/2803.

### Decision letter and Author response

Decision letter https://doi.org/10.7554/eLife.70792.sa1
Author response https://doi.org/10.7554/eLife.70792.sa2

## Additional files

### Supplementary files

• Transparent reporting form
• Source data 1. Source data on all displayed figures and graphs.

### Data availability

Raw data for the graphs and the western blot is provided in the Source Data files.

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
