## [Editor Report]

This paper shows that conditional knockout mice develop wide areas of loss of myelin integrity, as well as microglial inflammation within the affected areas. In association, a range of behavioral tests reveal abnormalities as the mice age. This work presents an important link between myelin loss and aging that will be of interest to a broad swathe of the scientific community.

---

## [Decision Letter]

**Decision letter after peer review:**

Thank you for submitting your article "Isolated catatonia-executive dysfunction in aged mice with forebrain-specific loss of myelin integrity" for consideration by *eLife*. Your article has been reviewed by 3 peer reviewers, and the evaluation has been overseen by a Reviewing Editor and Marianne Bronner as the Senior Editor. The following individuals involved in review of your submission have agreed to reveal their identity: Bruce Trapp (Reviewer #3).

Essential revisions:

There is general agreement that the study is well done, has several novel aspects that will be of interest to the field. The data convincingly shows evidence of pathology associated with forebrain and oligodendrocyte specific Plp1 mutant mice. However, there were several points and questions that need to be addressed prior to publication.

1. A more robust description of the pathological data would strengthen and enhance the impact of this paper.

– A more thorough and quantitative analysis of myelin in the mutant mice would strengthen the paper, as well as a more thorough histological analysis of changes in myelin integrity at earlier developmental time points to support to the authors' claims of aging-specific deficits.

A critical question that seems to be unanswered is whether PLP-negative oligos appropriately myelinate forebrain axons and maintain these myelin internodes. MBP staining could address this issue.

2. The overall impact would also be enhanced by a more global characterization of microglia activation. This could serve as a surrogate marker of the myelin alterations.

Previous work by the authors shows that, inhibiting inflammation using a drug (PLX5622) ameliorate phenotypes thought to be due directly to myelin loss. How do these findings relate? the paper would be stronger if this additional experiment had been performed to tease apart the direct and indirect effects of PLP loss (see reviewer comments below)

3. More histological analyses of younger ages are needed to support the conclusions (and title) and claim of age -related catatonia-executive dysfunction phenotype Multiple assessment timepoints are needed to identify the neurodegenerative onset r-elated to temporal changes in the PLP expression profile This could be addressed with more data to confirm the findings of aged-specific changes (see reviewer 1 comments) or please soften claims and change the title and emphases of age-related phenotype

*Reviewer #1 (Recommendations for the authors):*

To formulate a comprehensive understanding of higher brain functions that rely on motor output, we must probe these abilities in a manner that allows us to distinguish between deficits in forebrain from cerebellum and spinal cord myelin integrity. This work provides the field with a novel and important tool to begin to appreciate regional differences in myelination in relation to executive cognitive functioning that has broad implications in numerous neuropsychiatric disorders. While the study provides interesting evidence of pathology associated with forebrain and oligodendrocyte-specific Plp1 mutant mice, the work is largely descriptive and fails to provide in-depth insight regarding myelin microstructural changes, in particular assessment of myelin sheath thickness, compaction, or nodal changes. A more thorough histological analysis of changes in myelin integrity at earlier developmental time points would add a baseline comparison and further support to the authors' claims of aging-specific deficits. The authors speculate about the possible mechanisms that would explain the phenotypic changes associated with PLP1 depletion of forebrain oligodendrocytes, but they do not test any of these hypotheses. They identify markers of advanced aging in forebrain-specific and oligodendrocyte-specific Plp1 knockouts, such as increased APP spheroids and neuroinflammation, but do not provide a convincing causal relationship between these neuropathological findings and the subsequent deficiencies in myelin integrity.

– While the goal of the work is directly related to myelin structure and function, the authors fail to provide thorough and localized analysis of myelin in the mutant mice. Are there changes in the expression of other myelin proteins beside PLP and MBP? What is the g-ratio in these regions (using TEM)? The authors leave a great deal to speculation, especially as Figure 1 contains only qualitative data.

– The authors claim that Plp1 expression is restricted to oligodendrocytes and therefore, the phenotypic differences observed in the conditional knock out mice are caused by the PLP-deficient oligodendrocytes. However, in accordance with another paper by this group published in 2017 and research done by other groups, PLP is not only expressed by oligodendrocytes. Adding a control experiment with a higher magnification colocalization between a neuronal marker and PLP would be needed to assess PLP expression changes in forebrain neurons.

– The presence of neuroinflammation and neurodegenerative markers in mutant mice is interesting, but the authors fail to explain how the forebrain and oligodendrocyte-specific deficits in Plp1 are related to these findings? Is this a cause or consequence of changes in region-specific myelin?

– The authors emphasize the age-related phenotype and the generation of a new model to study myelin alteration specifically in aged mice, but they do not show any analysis of histology or neuropathology in young adults to claim the phenotype is only seen is aged mice. A histological analysis of young adult brains would be needed to compare phenotypes between ages and claim an age-dependent phenotype. Multiple assessment timepoints are needed here to identify the neurodegenerative onset related to temporal changes in the PLP expression profile.

– The title of the paper that emphasizes the role of age in the catatonia-executive dysfunction phenotype is not justified by the experiments and the results. First, the authors show catatonia dysfunction in young adults starting at 3 and 5 months in females and males, respectively (Figure 3A,B). Moreover, of the tests performed to evaluate executive function, only 1 was conducted in both young and aged adults (Figure 3E,F). In this particular executive test in which the effect of the age is analyzed, the authors show that males present executive dysfunction when aged, but females exhibit deficit at 6 months but not 20 months. I suggest changing the title and eliminating the emphases of age-related phenotype unless a consistent evaluation and finding of aged-specific changes can be confirmed.

– I greatly appreciate that the authors broke down behavior based on biological sex and applaud them for taking this important variable into consideration. The authors could strengthen their analyses by including this variable in their histological assessment as they have in their behavioral analyses. As is, the paper shows no analysis of neuropathology or histology in females, even if they indicate in statistical methodology that mice of both sexes were used. Considering Plp is an X-linked gene and that the authors show sexual dimorphism in executive function, adding histological analysis in females would be needed to reach the author's conclusions.

– As a point of consideration, changes in motor function have been identified in a model of increased myelination specific to the cortico-callosal projections in the corpus callosum and not the cortico-fugal projections (Gibson et al. 2014). As one of the key points of this manuscript is the development of a model in which changes in myelination occur only in the forebrain, including the corpus callosum, in order to isolate executive functioning from motor functioning, how does one reconcile that changes in motor function have been characterized with only changes in the corpus callosum and not cerebellum or spinal cord regions as identified in Gibson et al. 2014. This may be worth discussing.

*Reviewer #2 (Recommendations for the authors):*

The results are clearly presented and the experimental strategy is powerful. The inclusion of data on the complex running wheel used by other authors to study the effects of loss of oligodendrocyte generation was a nice addition, as the lack of phenotype on the wheel due to the conditional knockout enables an interesting comparison with the results of other laboratories.

A significant weakness, though, is that the authors have not distinguished the effects of loss of myelin integrity and inflammation. As they showed beautifully in a recent JCI paper, inhibiting inflammation using a drug (PLX5622) administered in the chow can ameliorate phenotypes thought to be due directly to myelin loss. Although the results presented here are very important and represent a significant step in our understanding of myelin functions, the paper would certainly be stronger if this additional experiment had been performed so as to tease apart the direct and indirect effects of PLP loss.

*Reviewer #3 (Recommendations for the authors):*

Myelin-axon alterations can cause neuronal/axonal dysfunction and neurological disability. The best examples of this are rodent studies that alter myelin protein content without causing demyelination. The best studied example is deletion of the major CNS myelin protein proteolipid protein (PLP). PLP deletions cause a late onset neurodegeneration modelling human spastic paraplegia type 2. Studies of the global PLP deletion has focused on motor disability. These and other studies have led to the concept that myelin maintains axonal integrity. This manuscript asks the question of whether loss of myelin integrity in the forebrain has behavioral consequences. PLP was selectively removed from ventricular zone stem cells of the mouse forebrain. This resulted in an absence of PLP in oligodendrocytes in the prefrontal cortex, corpus callosum and hippocampal fimbria. This data is convincing and well-documented. These mice showed normal motor function and no evidence of spasticity. In contrast, the mice showed late onset deficits in executive function and displayed catatonic-like symptoms. There are two major strengths to this paper. The site specific deletion of PLP from ventricular zone stem cells using the EMZ promotor and the sophisticated behavioral testing.

A major impact of this paper relate to the role of myelin-axon integrity in ageing. Subtle changes in cortical myelin have been described in the aged primate cortex. MRI abnormalities have also been found in myelinated periventricular white matter in individuals with mild cognitive impairment and individuals with multiple sclerosis. The concept that the axonal dysfunction can be caused by subtle changes in myelin composition therefore are likely to be of clinical importance.

Pathological characterization was sufficient to support the author's conclusions that axonal integrity is compromised in CNS regions that harbor the EMZ driven PLP KO. The use of light sheet microscopy to chart the distribution of PLP-positive oligodendrocytes is impressive. Can some comment be made about the consistency of this data? How many mice were examined? The pathological data could be enhanced by thicker sections and wider views of the tissue. It would be nice to see how increased GFAP or microglial activation markers correspond to light sheet images of PLP-null oligodendrocytes.

This is a clearly written manuscript and the data supports the conclusions of the authors. It is among the first papers to report manipulation of a subpopulation of oligodendrocytes. It also provides an animal model with cortical oligodendrocyte and myelin abnormalities in the absence of demyelination. This has implication to axonal degeneration in ageing, MCI and MS. The model and behavioral data are described in detail. A more robust description of the pathological data would enhance the impact of this paper. What is the distribution of activated microglia? Do they reflect the distribution of PLP-null oligodendrocytes? There are other fundamental questions that will need to be addressed. Does the transgene alter the production of other cell types? What causes the apparent atrophy (ventricular enlargement) that was described as obvious? Despite these limitations, the experimental proof-of-concept approach opens an array of possibilities for futures and more detailed studies.

---

## [Author Response]

Essential revisions:There is general agreement that the study is well done, has several novel aspects that will be of interest to the field. The data convincingly shows evidence of pathology associated with forebrain and oligodendrocyte specific Plp1 mutant mice. However, there were several points and questions that need to be addressed prior to publication.

We thank the reviewing editor and the reviewers for their constructive comments on our study, and for the invitation to submit a revised manuscript. We have addressed all of the reviewers' comments in a detailed point-to-point response below. In the revised version of the manuscript, new data, figure adaptations and remodeled text have been highlighted in blue. We also took the opportunity to remodel several figures to increase clarity. We believe that this revision process has greatly strengthened our manuscript.

There has been a significant Covid-related delay in preparing this revision. We confirm that no other publication has appeared in the meantime that would jeopardize the novelty of our findings.

1. A more robust description of the pathological data would strengthen and enhance the impact of this paper.– A more thorough and quantitative analysis of myelin in the mutant mice would strengthen the paper, as well as a more thorough histological analysis of changes in myelin integrity at earlier developmental time points to support to the authors' claims of aging-specific deficits.A critical question that seems to be unanswered is whether PLP-negative oligos appropriately myelinate forebrain axons and maintain these myelin internodes. MBP staining could address this issue.

We agree and have performed, as suggested, a quantitative analysis of myelin changes in the forebrain of PLP cKO mice by immunostaining – in addition to the more ‘qualitative’ light sheet microscopy (New Figure 1 A and A’). We quantified the areas with PLP-positive myelin in the cortex and in the corpus callosum, both recombination sites of *Emx^IRESCre^* , and for comparison the striatum that should not be affected by Cre recombination. Expectedly, we detected a robust decrease in PLP-positive area in the cKO animals. The reduction was most striking in the cortex. In the ventral corpus callosum significant amounts of PLP+ myelin could still be detected. We suspect that this myelin – rather than being the result of insufficient Cre recombination – originates from OPC/oligodendrocytes migrating into the v entral corpus callosum from the ‘non-recombined’ striatum. We present and discuss these points in the revised version of the manuscript.

We also performed a quantitation by western blots for MBP and PLP in the prefrontal cortex (recombined region) and in the cerebellum (non-recombined) region (New Figure 1D, quantification panels). As suggested by the reviewers, we also performed PLP and MBP co-labelling to determine whether forebrain axons are properly myelinated. This revealed a significant reduction also of MBP positive profiles (New Figure 1 – supplement 1), especially in the upper cortical layers where myelination is already sparse. It suggests that in the absence of PLP myelinated internodes are either lost over time or not properly formed during development.

We refrained from a detailed ultrastructural analysis of PLP-deficient myelin as this has been extensively done and published in the past (Klugmann et al., 1997; Möbius et al. 2016; Patzig et al. 2016; Rosenbluth et al., 2006, Steyer et al. 2020; Trevisiol et al. 2021).

Our claim of “aging-specific defects” reflects the fact that pathology and behavioral anomalies in PLP-deficient mice manifest with increasing age (as for example in the hurdle test of male mice). As seen in conventional PLP KO mice, the associated phenotype is of progressive nature and worsens in aging mice. However, we agree with the reviewers that the experimental base for this claim in the present paper is weak, as we did not perform every behavioral test at multiple time points. We have therefore removed these claims from the revised version of the manuscript.

2. The overall impact would also be enhanced by a more global characterization of microglia activation. This could serve as a surrogate marker of the myelin alterations.Previous work by the authors shows that, inhibiting inflammation using a drug (PLX5622) ameliorate phenotypes thought to be due directly to myelin loss. How do these findings relate? the paper would be stronger if this additional experiment had been performed to tease apart the direct and indirect effects of PLP loss (see reviewer comments below)

We thank the reviewers for this suggestion. We now have performed the global characterisation of gliosis in the forebrain using immunofluorescence staining of coronal brain slices (New Figure 2A,B). We also performed a quantitative analysis focusing on the cortex and corpus callosum (both Cre-recombined areas) in comparison to the striatum as a (non-recombined) control region. Robust microgliosis (Iba1+ area) and astrogliosis (GFAP+ area) was only seen in the recombined white matter (corpus callosum). This is in agreement with Figure 2 – supplement 1 in which we quantified areas immunopositive for microglia and astrocyte markers in the cortex and corpus callosum using immunohistochemistry with chromogenic labeling.

As correctly pointed out by the reviewer (and cited in our manuscript), we had previously performed microglia depletion experiments in CnpKO animals, another model of myelin dysfunction (Janova et al., JCI 2018). In these mutants, a novel catatonia-like behavioral phenotype was detected that was apparently driven by secondary microgliosis (there is no obvious myelin loss). And it is highly probable that the shared catatonia-like phenotype in any Plp null mutant can likewise be ameliorated by microglial depletion. However, we feel that duplicating the entire experimental series in the presence of PLX5622 would exceed the scope of this paper. This is because the aim of the present study was to explore in how far forebrain-specific myelin perturbations can define behavioral consequences, not what the direct or indirect mechanisms are.

3. More histological analyses of younger ages are needed to support the conclusions (and title) and claim of age -related catatonia-executive dysfunction phenotype Multiple assessment timepoints are needed to identify the neurodegenerative onset r-elated to temporal changes in the PLP expression profile This could be addressed with more data to confirm the findings of aged-specific changes (see reviewer 1 comments) or please soften claims and change the title and emphases of age-related phenotype

We completely agree and as discussed in more detail above (#1), we have softened any claim of “aging-specific” effects and have removed this term from the title. Moreover, the revised manuscript no longer emphasizes an age-dependent phenotype.

Reviewer #1 (Recommendations for the authors):To formulate a comprehensive understanding of higher brain functions that rely on motor output, we must probe these abilities in a manner that allows us to distinguish between deficits in forebrain from cerebellum and spinal cord myelin integrity. This work provides the field with a novel and important tool to begin to appreciate regional differences in myelination in relation to executive cognitive functioning that has broad implications in numerous neuropsychiatric disorders. While the study provides interesting evidence of pathology associated with forebrain and oligodendrocyte-specific Plp1 mutant mice, the work is largely descriptive and fails to provide in-depth insight regarding myelin microstructural changes, in particular assessment of myelin sheath thickness, compaction, or nodal changes. A more thorough histological analysis of changes in myelin integrity at earlier developmental time points would add a baseline comparison and further support to the authors' claims of aging-specific deficits. The authors speculate about the possible mechanisms that would explain the phenotypic changes associated with PLP1 depletion of forebrain oligodendrocytes, but they do not test any of these hypotheses. They identify markers of advanced aging in forebrain-specific and oligodendrocyte-specific Plp1 knockouts, such as increased APP spheroids and neuroinflammation, but do not provide a convincing causal relationship between these neuropathological findings and the subsequent deficiencies in myelin integrity.– While the goal of the work is directly related to myelin structure and function, the authors fail to provide thorough and localized analysis of myelin in the mutant mice. Are there changes in the expression of other myelin proteins beside PLP and MBP? What is the g-ratio in these regions (using TEM)? The authors leave a great deal to speculation, especially as Figure 1 contains only qualitative data.

The primary objective of this study was to investigate the effects of forebrain-restricted myelin dysfunctions on behavioural tasks that are related to higher cognitive functions. We therefore refrained from any detailed analysis of myelin integrity (TEM analysis, g-ratios etc) given the large number of prior publications by us and others describing the myelin phenotype of PLP null mutant mice (Klugmann et al., 1997; Möbius et al. 2016; Patzig et al. 2016; Rosenbluth et al., 2006, Steyer et al. 2020; Trevisiol et al. 2021). We have no reason to believe that PLP-deficient myelin in forebrain-specific KO mice is qualitatively different from the conventional null allele, and have referred to these earlier studies. Moreover, we have now provided quantification of MBP and PLP immunostainings (New Figure 1 and New Figure 2) and quantifications of western blot (New Figure 1D).

– The authors claim that Plp1 expression is restricted to oligodendrocytes and therefore, the phenotypic differences observed in the conditional knock out mice are caused by the PLP-deficient oligodendrocytes. However, in accordance with another paper by this group published in 2017 and research done by other groups, PLP is not only expressed by oligodendrocytes. Adding a control experiment with a higher magnification colocalization between a neuronal marker and PLP would be needed to assess PLP expression changes in forebrain neurons.

Neurons and other cells can indeed show trace amounts of Plp1 RNA transcripts. We had therefore conducted an earlier study to specifically investigate the neuropathological consequences of neuronal loss of Plp expression and its possible contribution to the pathology seen in conventional Plp null mice. To this end, we created *Nex-Cre::Plp^fl/fl^* mice which recombine only in excitatory neurons of the forebrain (Lueders et al. 2017, PMID: 28836307). We failed to detect any axonopathy or gliosis, rendering the phenotype of *Emx^IRESCre^ Plp^fl/fl^* very unlikely to be caused by neuronal PLP. We think that this prior work which we cited in the current manuscript is convincing evidence against relevance of deleting neuronal PLP expression in the emergence of neuropathology. Moreover, PLP immuno-labeling in our hands (Knockout-validated aa3 anti-PLP antibody) does not show evidence of PLP protein expression in neurons but exclusively expression in myelinated fibers and oligodendrocytes (as shown in Author response image 1, with layer 5 neurons as black ‘holes’ upon PLP immunostaining).

**Author response image 1. sa2fig1:** 

– The presence of neuroinflammation and neurodegenerative markers in mutant mice is interesting, but the authors fail to explain how the forebrain and oligodendrocyte-specific deficits in Plp1 are related to these findings? Is this a cause or consequence of changes in region-specific myelin?

By genetically inactivating the expression of a myelin protein, one can be certain that neuroinflammation and neurodegeneration are secondary effects, caused by a primary absence of a myelin protein. Even an oligodendrocyte-specific *Plp1* null mutant (Lueders et al. 2017) recapitulates all of the phenotypes described in constitutive *Plp1* KOs (Klugmann et al., 1997; Griffiths et al., 1998). Thus, the downstream effects of PLP deficiency are secondary to a primary oligodendrocyte/myelin-linked dysfunction.

– The authors emphasize the age-related phenotype and the generation of a new model to study myelin alteration specifically in aged mice, but they do not show any analysis of histology or neuropathology in young adults to claim the phenotype is only seen is aged mice. A histological analysis of young adult brains would be needed to compare phenotypes between ages and claim an age-dependent phenotype. Multiple assessment timepoints are needed here to identify the neurodegenerative onset related to temporal changes in the PLP expression profile.

We thank the reviewer for this suggestion and have investigated by immunofluorescence gliosis also at an earlier time point, i.e.11 months of age (Figure 2). Importantly, in this younger cohort we also detected significant gliosis of the corpus callosum. Gliosis in this mutant is thus not dependent on advanced brain aging although a progression of this phenotype is very likely. Based on these 11 months data, we have removed the qualifier “age-dependent” from the revised manuscript.

– The title of the paper that emphasizes the role of age in the catatonia-executive dysfunction phenotype is not justified by the experiments and the results. First, the authors show catatonia dysfunction in young adults starting at 3 and 5 months in females and males, respectively (Figure 3A,B). Moreover, of the tests performed to evaluate executive function, only 1 was conducted in both young and aged adults (Figure 3E,F). In this particular executive test in which the effect of the age is analyzed, the authors show that males present executive dysfunction when aged, but females exhibit deficit at 6 months but not 20 months. I suggest changing the title and eliminating the emphases of age-related phenotype unless a consistent evaluation and finding of aged-specific changes can be confirmed.

We agree with the reviewer (see also above) and have removed this phrase from the title, and also softened claims related to age throughout the manuscript.

– I greatly appreciate that the authors broke down behavior based on biological sex and applaud them for taking this important variable into consideration. The authors could strengthen their analyses by including this variable in their histological assessment as they have in their behavioral analyses. As is, the paper shows no analysis of neuropathology or histology in females, even if they indicate in statistical methodology that mice of both sexes were used. Considering Plp is an X-linked gene and that the authors show sexual dimorphism in executive function, adding histological analysis in females would be needed to reach the author's conclusions.

As described in the essential revisions section above: in our analysis of gliotic reactions and APP+ swellings, we had separately investigated female and male mice, but had merged these data for presentation and quantification. We now present these findings in a sex-specific manner (Figure 2 – supplement 1). Importantly, this did not show any difference for male and female forebrains of PLP cKO mice. We also take this as an indication that the underlying myelin pathology (and number of recombined oligodendrocytes) is the same in male and female mutant mice.

– As a point of consideration, changes in motor function have been identified in a model of increased myelination specific to the cortico-callosal projections in the corpus callosum and not the cortico-fugal projections (Gibson et al. 2014). As one of the key points of this manuscript is the development of a model in which changes in myelination occur only in the forebrain, including the corpus callosum, in order to isolate executive functioning from motor functioning, how does one reconcile that changes in motor function have been characterized with only changes in the corpus callosum and not cerebellum or spinal cord regions as identified in Gibson et al. 2014. This may be worth discussing.

We thank the reviewer for pointing this out. We are aware that motor cortical areas are involved in motor learning skills, which is why we had performed the complex running wheel analysis. Rather than isolating motor phenotypes from executive functioning, the aim of our genetic targeting approach was to uncouple any forebrain-specific (motor) functions from major motor dysfunctions observed in Plp null mice that reflect axonal loss in cerebellum and spinal cord (Griffiths et al., 1998). In our view, the findings by Gibson et al.2014 are, therefore, not in conflict with the data presented in this paper. We have added a respective sentence to the discussion.

Reviewer #2 (Recommendations for the authors):The results are clearly presented and the experimental strategy is powerful. The inclusion of data on the complex running wheel used by other authors to study the effects of loss of oligodendrocyte generation was a nice addition, as the lack of phenotype on the wheel due to the conditional knockout enables an interesting comparison with the results of other laboratories.

We thank the reviewer for his/her positive comments on our manuscript.

A significant weakness, though, is that the authors have not distinguished the effects of loss of myelin integrity and inflammation. As they showed beautifully in a recent JCI paper, inhibiting inflammation using a drug (PLX5622) administered in the chow can ameliorate phenotypes thought to be due directly to myelin loss. Although the results presented here are very important and represent a significant step in our understanding of myelin functions, the paper would certainly be stronger if this additional experiment had been performed so as to tease apart the direct and indirect effects of PLP loss.

See also Reviewer 1, point #2. As correctly pointed out by this reviewer (and cited in our manuscript), we had previously performed microglia depletion experiments in Cnp KO animals, another model of myelin dysfunction (Janova et al., JCI 2018). In these mutants, a novel catatonia-like behavioral phenotype was detected that was driven by secondary microgliosis. It is indeed highly likely that also the shared catatonia-like phenotype of Plp null mutant mice can be ameliorated by microglial depletion.

However, we feel that duplicating the experimental series in the presence of PLX5622 would exceed the scope of this paper. Most of all, the aim of the present study was to explore in how far forebrain-specific myelin perturbations can cause behavioral consequences in mice (not what the direct and indirect downstream mechanisms are that lead to altered neuronal functions).

We certainly agree that here the role of secondary microgliosis remains a very interesting aspect that we wish to address in the future, presumably first in conventional PLP KO mice, in which the interaction of altered microglia with the synaptic circuitry needs to be dissected at various levels. Again, we think that the necessary experiments exceed by far the scope of the current paper.

Reviewer #3 (Recommendations for the authors):Myelin-axon alterations can cause neuronal/axonal dysfunction and neurological disability. The best examples of this are rodent studies that alter myelin protein content without causing demyelination. The best studied example is deletion of the major CNS myelin protein proteolipid protein (PLP). PLP deletions cause a late onset neurodegeneration modelling human spastic paraplegia type 2. Studies of the global PLP deletion has focused on motor disability. These and other studies have led to the concept that myelin maintains axonal integrity. This manuscript asks the question of whether loss of myelin integrity in the forebrain has behavioral consequences. PLP was selectively removed from ventricular zone stem cells of the mouse forebrain. This resulted in an absence of PLP in oligodendrocytes in the prefrontal cortex, corpus callosum and hippocampal fimbria. This data is convincing and well-documented. These mice showed normal motor function and no evidence of spasticity. In contrast, the mice showed late onset deficits in executive function and displayed catatonic-like symptoms. There are two major strengths to this paper. The site specific deletion of PLP from ventricular zone stem cells using the EMZ promotor and the sophisticated behavioral testing.A major impact of this paper relate to the role of myelin-axon integrity in ageing. Subtle changes in cortical myelin have been described in the aged primate cortex. MRI abnormalities have also been found in myelinated periventricular white matter in individuals with mild cognitive impairment and individuals with multiple sclerosis. The concept that the axonal dysfunction can be caused by subtle changes in myelin composition therefore are likely to be of clinical importance.Pathological characterization was sufficient to support the author's conclusions that axonal integrity is compromised in CNS regions that harbor the EMZ driven PLP KO. The use of light sheet microscopy to chart the distribution of PLP-positive oligodendrocytes is impressive. Can some comment be made about the consistency of this data? How many mice were examined? The pathological data could be enhanced by thicker sections and wider views of the tissue. It would be nice to see how increased GFAP or microglial activation markers correspond to light sheet images of PLP-null oligodendrocytes.

We thank the reviewer for commending our light sheet microscopic analysis. We agree that the LSM analysis provides a great overall impression of recombination territories in forebrain PLP cKO. Quantifications of LSM can be challenging due to limited antibody penetration, which is why we do not provide quantifications of these datasets. However, in the revised version of main Figure 1 we now provide 2D quantifications of PLP stainings in coronal paraffin slices.

This is a clearly written manuscript and the data supports the conclusions of the authors. It is among the first papers to report manipulation of a subpopulation of oligodendrocytes. It also provides an animal model with cortical oligodendrocyte and myelin abnormalities in the absence of demyelination. This has implication to axonal degeneration in ageing, MCI and MS. The model and behavioral data are described in detail. A more robust description of the pathological data would enhance the impact of this paper. What is the distribution of activated microglia? Do they reflect the distribution of PLP-null oligodendrocytes? There are other fundamental questions that will need to be addressed. Does the transgene alter the production of other cell types? What causes the apparent atrophy (ventricular enlargement) that was described as obvious? Despite these limitations, the experimental proof-of-concept approach opens an array of possibilities for futures and more detailed studies.

We thank the reviewer for the positive evaluation of our manuscript. In the revised version of main Figure 2, we now show overview images of micro- and astrogliosis that should give a better impression of the global distribution of these cell populations. We also included a control area in which Emx-Cre is not expressed during development (striatum). Indeed, glia activation is only seen in brain areas affected by recombination. Interestingly, we could only find significantly increased neuroinflammation in the white matter (corpus callosum) but not in the cortical grey matter although this region is clearly recombined in Emx-Cre-positive animals. Presumably, microglia activation only manifests to the extent that it becomes detectable by immunohistochemistry in heavily myelinated areas, which might also be most affected by myelin dysfunction.

We have addressed the other points in the response to the essential revisions.